# FlexWorld: Progressively Expanding 3D Scenes for Flexible-View Exploration

**Luxi Chen**[1,2,3,*] **Zihan Zhou**[1,2,3*]**, Min Zhao**[4]**, Yikai Wang**[5†]**, Ge Zhang**[6]**,**
**Wenhao Huang**[6]**, Hao Sun**[1,2,3]**, Ji-Rong Wen**[1,2,3]**, Chongxuan Li**[1,2,3†]
[1] Gaoling School of Artificial Intelligence, Renmin University of China
[2] Beijing Key Laboratory of Research on Large Models and Intelligent Governance
[3] Engineering Research Center of Next-Generation Intelligent Search and Recommendation, MOE
[4] Dept. of Comp. Sci. & Tech., BNRist Center, THU-Bosch MLCenter, Tsinghua University
[5] School of Artificial Intelligence, Beijing Normal University
[6] ByteDance Seed

## Abstract

Generating flexible-view 3D scenes, including 360° rotation and zooming, from single images is challenging due to a lack of 3D data. To this end, we introduce FlexWorld, a novel framework that progressively constructs a persistent 3D Gaussian splatting representation by synthesizing and integrating new 3D content. To handle novel view synthesis under large camera variations, we leverage an advanced pre-trained video model fine-tuned on accurate depth-estimated training pairs. By combining geometry-aware scene integration and optimization, Flex-World refines the scene representation, producing visually consistent 3D scenes with flexible viewpoints. Extensive experiments demonstrate the effectiveness of FlexWorld in generating high-quality novel view videos and flexible-view 3D scenes from single images, achieving superior visual quality under multiple popular metrics and datasets compared to existing state-of-the-art methods. Additionally, FlexWorld supports extrapolating from existing 3D scenes, further extending its applicability. Qualitatively, we highlight that FlexWorld can generate high-fidelity scenes that enable 360° rotations and zooming exploration. Our code is available at `https://github.com/ML-GSAI/FlexWorld`.

## 1  Introduction

Creating a 3D scene with flexible views from a single image holds transformative potential for applications where direct 3D data acquisition is costly or impractical, such as archaeological preservation and autonomous navigation. However, this task remains fundamentally ill-posed: a single 2D observation provides insufficient information to disambiguate the complete 3D structure. In particular, when extrapolating to extreme viewpoints (e.g., 180° rotations), previously occluded or entirely absent content may emerge, introducing significant *uncertainty*.

Generative models, particularly diffusion models [1, 2, 3], offer a principled and effective solution to this problem. While existing methods often rely on pre-trained generative models as priors for novel view synthesis, they face notable limitations. Image-based diffusion methods [4, 5, 6, 7] tend to accumulate geometric errors, whereas video-based diffusion approaches [8, 9] struggle with dynamic content and poor camera supervision. Recent attempts [10, 11] to incorporate point cloud priors for

---

*Equal Contribution.
†Correspondence to Chongxuan Li <chongxuanli@ruc.edu.cn>, Yikai Wang <yikaiw@bnu.edu.cn>.

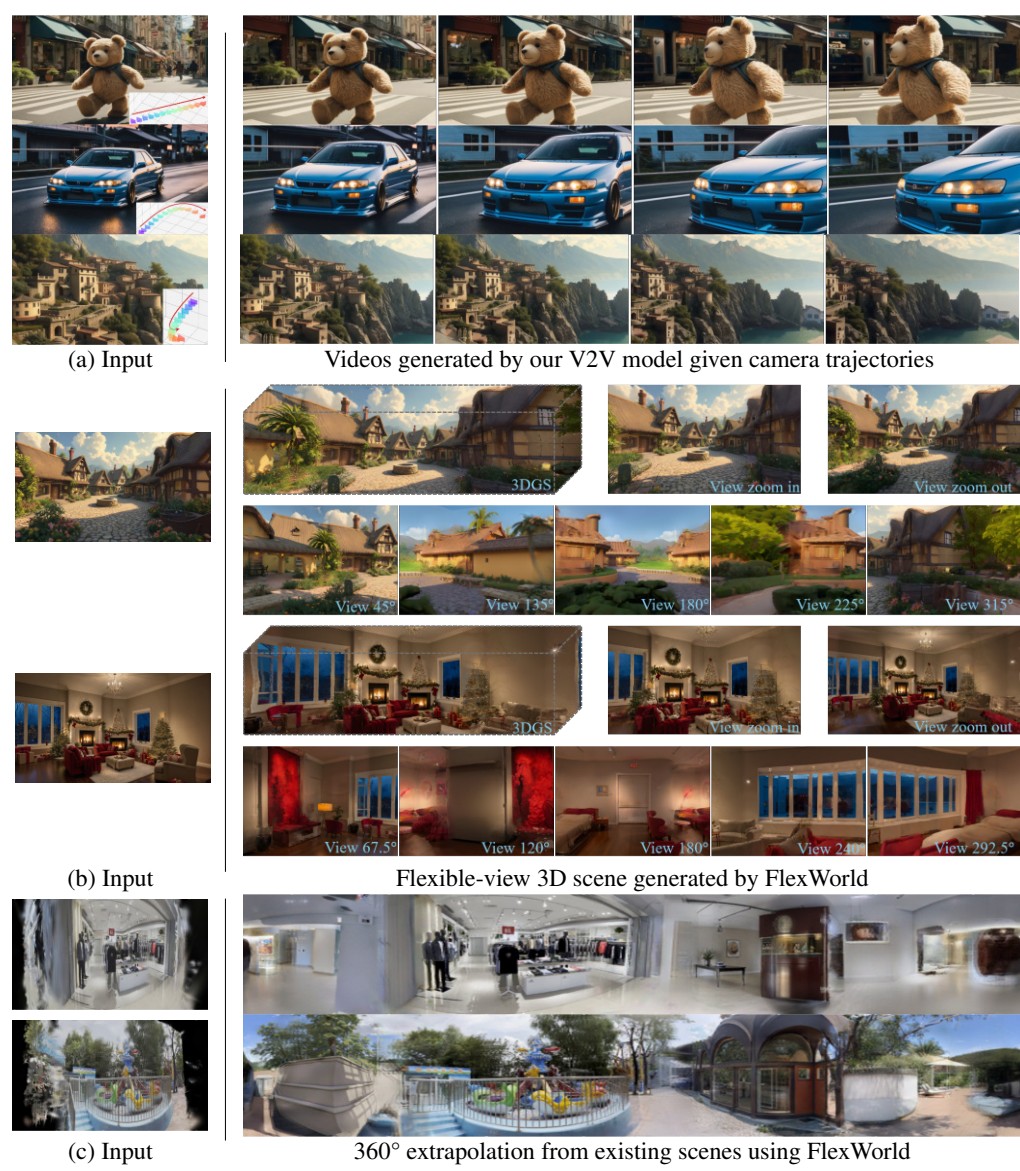

(a) Input      Videos generated by our V2V model given camera trajectories

(b) Input      Flexible-view 3D scene generated by FlexWorld

(c) Input      360° extrapolation from existing scenes using FlexWorld

Figure 1: **FlexWorld generates high-quality videos with camera control and flexible-view 3D scenes progressively.** (a) FlexWorld introduces a V2V diffusion producing high-quality videos from incomplete scene renderings given diverse camera trajectories with *large variation*. (b) FlexWorld progressively generates *flexible-views* (e.g., 360° rotations and zooming) 3DGS scenes via the V2V model. (c) FlexWorld further supports extrapolating an existing scene into 360° exploration.

improved consistency have shown promise but remain limited in scalability, often failing under large viewpoint changes, further limiting the flexibility in exploring generated 3D scenes.

To this end, we propose FlexWorld for flexible-view 3D scene generation from single images. In contrast to existing methods [12, 13, 14], FlexWorld maintains a *persistent* 3D Gaussian splatting representation that is *progressively expanded* by synthesizing and integrating novel 3D content. To ensure robust novel view synthesis under large camera variations, we develop a video-to-video (V2V) model with an advanced video foundation model [15] fine-tuned on accurate depth-estimated training pairs. The V2V model generates complete view images from incomplete ones rendered from the current scene. Through geometry-aware scene integration and optimization, FlexWorld progressively details the persistent scene representation, ultimately producing flexible-view 3D scenes with strong geometric and visual consistency from single images and extrapolating from existing scenes.

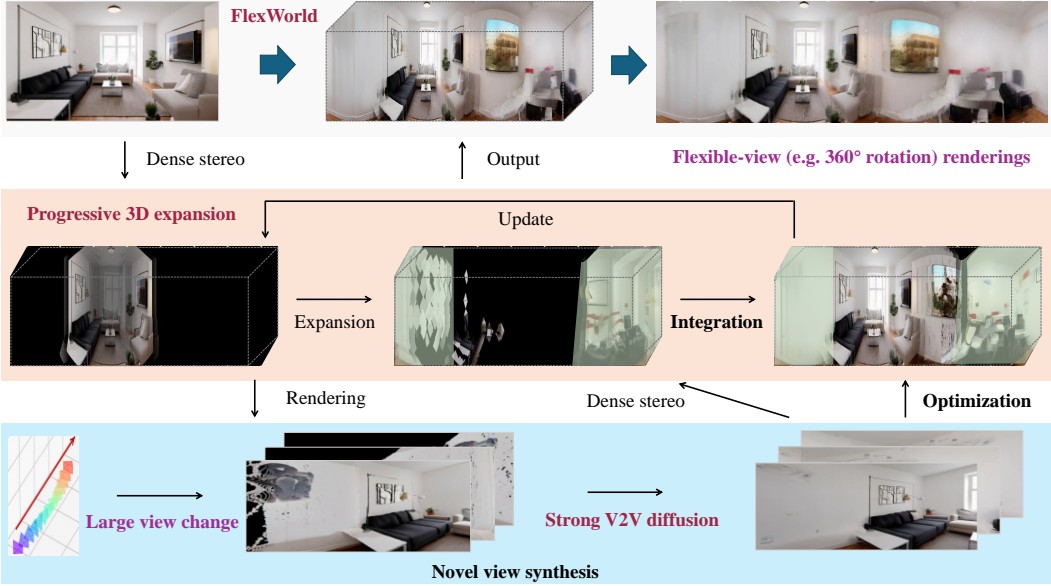

Figure 2: **Overview of FlexWorld.** FlexWorld trains a strong V2V diffusion capable of generating high-quality videos from incomplete views rendered from coarse 3D scenes. It progressively expands the 3D scene by adding new 3D content estimated from the refined videos via scene integration. Ultimately, from a single image, it yields a detailed 3D scene capable of rendering flexible viewpoints.

Our extensive experiments demonstrate FlexWorld's effectiveness in both high-quality video and flexible-view 3D scene synthesis. In particular, our V2V model achieves superior visual quality compared to the current state-of-the-art baselines [8, 9, 13, 11, 10] while maintaining excellent camera controllability across multiple benchmarks [16, 17] (see Tab. 1). A similar conclusion holds for the 3D scene generation benchmarks (see Tab. 2). In addition, FlexWorld enables the synthesis of the 3D scene with flexible view in high-fidelity and the extrapolation of existing DL3DV [18] scenes (see Fig. 1) into 360° exploration, consistent with our quantitative results.

In summary, our key contributions are:

- We propose FlexWorld, a progressive framework for flexible-view 3D scene generation that builds and refines a persistent scene representation by expanding it with synthesized novel views.
- We introduce a video-to-video diffusion model fine-tuned on optimized training data under an advanced base model, enabling consistent novel view synthesis under large camera variations.
- FlexWorld exhibits superior performance in video and scene generation compared with baseline models on various benchmark datasets [16, 17] while supporting scene extrapolation.

## 2 Related work

### 2.1 3D scene generation

With the emergence of 3D representations that enable differentiable rendering [19, 20], 3D object generation from single texts/images has advanced rapidly [21, 22, 23, 24, 25, 26, 27, 28, 29], closely followed by advancements in 3D scene generation. Several works [4, 30, 6, 7, 5, 31, 32] employ image diffusion models [33, 34] for novel view synthesis and monocular depth estimation [35, 36, 37, 38] to derive 3D structures for corresponding views. Another line of the work [39, 40, 41, 42, 43, 44, 45] involves training a network to obtain a 3D representation from single or sparse images directly. Recent studies have integrated camera control into image [46, 47, 48, 11] or video models [12, 14, 10, 13, 49] to facilitate novel view synthesis, subsequently performing 3D reconstruction [50, 51] to obtain representations of 3D scenes. The method using a video model can generate scenes with better consistency. However, constrained by large viewpoint changes in videos generated in a single pass and the neglect of integrating generated videos into existing 3D scenes, these methods ultimately

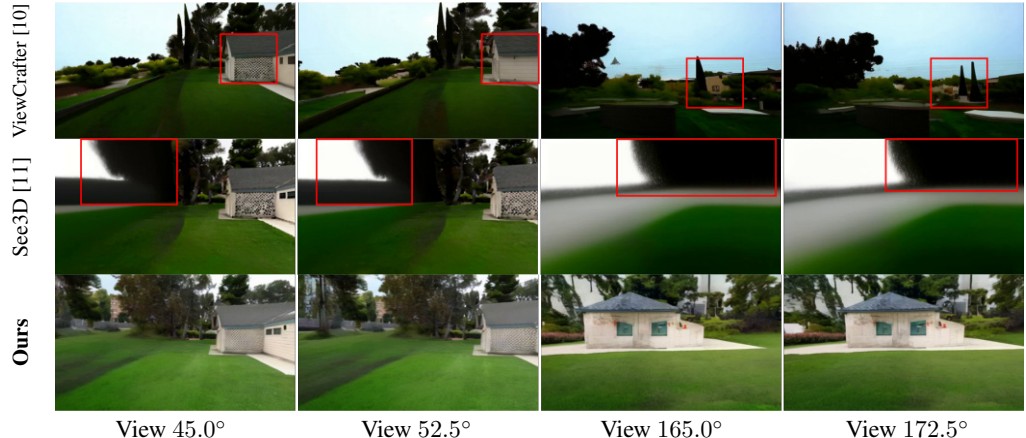

Figure 3: **We improve our video diffusion model to enable generating 3D consistent videos under large camera variation.** We present novel views generated from each model when the camera is rotated 180 degrees to the left. The red bounding box indicates 3D inconsistency or poor visual quality in the generated content. Our model generates higher quality and more consistent 3D scenes.

result in limited scene scale. Leveraging a persistent 3DGS representation and a strong V2V capable of handling large view variations, FlexWorld progressively expands the viewable area, ultimately enabling flexible-view 3D scenes generation and scene extrapolation.

## 2.2 Camera-controlled video diffusion models

Recently, Camera-controlled video diffusion models have received widespread attention. Several works [8, 9, 52, 53, 54] explore the generation of videos under camera conditions. However, these models are not designed for static scene generation, as the dynamics in the generated videos hinder reconstruction. DimensionX [13] achieves basic camera control via several LoRAs [55] but lacks flexibility in complex movements. Wonderland [12] and StarGen [14] can generate videos from single views and camera trajectories; however, they are unable to produce new videos to complement existing 3D structures, restricting the range of generated scenes. See3D [11] and ViewCrafter [10] can accept missing scene information from specific cameras and perform completion, but they struggle to accommodate significant viewpoint changes (see Fig. 3). In contrast, we propose training a V2V model on a more advanced video foundation model, leveraging existing scene information to enable large camera variation and offering a powerful tool for flexible-view 3D scene generation.

## 3 Method

In this section, we will first introduce the preliminaries for FlexWorld. Subsequently, we will present our flexible-view 3D scene generation framework in Sec. 3.2, where our improved V2V model that supports our framework will be further discussed in Sec. 3.3.

## 3.1 Preliminaries

**Video diffusion model.** A diffusion model [1, 2, 3] consists of a forward and a denoising process. In the forward process, the diffusion model gradually adds Gaussian noise to a clean image $x_0$ from time 0 to $T$. The noisy image $x_t$ at a certain time $t \in [0, T]$ can be expressed as $x_t = \alpha_t x_0 + \sigma_t \epsilon$, where $\alpha_t$ and $\sigma_t$ are predefined hyperparameters. In the denoising process, a noise predictor $\epsilon_\theta(x_t, t)$ with parameters $\theta$ is trained to predict noise in $x_t$ for generation. Given the corresponding condition $y$ for $x$, the training objective of a diffusion model is:

$$\min_\theta \mathbb{E}_{t \sim \mathcal{U}(0,1), \epsilon \sim \mathcal{N}(0,I)} \left[ \| \epsilon_\theta(x_t, t; y) - \epsilon \|_2^2 \right]. \tag{1}$$

Recent video diffusion models [56, 57, 58, 15, 59, 60] typically employ a 3D-VAE [61] encoder $\mathcal{E}$ to compress the source video into a latent space where the diffusion model is trained. The generated latent video is subsequently decoded to the pixel space using the corresponding decoder $\mathcal{D}$.

**Dense stereo model.** The dense stereo models [50, 51, 62], e.g., DUSt3R [50] and MASt3R [51], provide an advanced tool for obtaining corresponding point maps, depth maps, and camera parameters from single or sparse views, facilitating the reconstruction of 3D point clouds. This approach offers a means to derive a coarse 3D structure and camera estimation from a single image.

**3D Gaussian splatting.** As a leading 3D representations, 3D Gaussian splatting (3DGS) [20] models the 3D scene by multiple 3D Gaussians parameterized by colors, centers, opacities, scales, and rotation quaternions. The effectiveness and efficiency of 3DGS in 3D reconstruction and generation have been widely demonstrated [20, 45, 44, 26, 27, 29]. In addition to the $\mathcal{L}_1$ loss and SSIM loss $\mathcal{L}_{\text{SSIM}}$ [63] presented in the original paper [20], optimizing a 3D scene's loss function typically incorporates the LPIPS loss $\mathcal{L}_{\text{LPIPS}}$ [64, 13] to improve optimization. The weights $\lambda_1$, $\lambda_{\text{SSIM}}$, and $\lambda_{\text{LPIPS}}$ are adjustable hyperparameters. Formally, the specific loss function is expressed as:

$$\mathcal{L} = \lambda_1 \mathcal{L}_1 + \lambda_{\text{SSIM}} \mathcal{L}_{\text{SSIM}} + \lambda_{\text{LPIPS}} \mathcal{L}_{\text{LPIPS}}. \tag{2}$$

### 3.2 Progressive scene expansion with persistent representation

To overcome the limitation of insufficient multi-views in single videos for continuous and flexible-view 3D scene generation (as discussed in Sec. 2.1), we propose FlexWorld, which addresses this challenge by maintaining a persistent 3DGS representation. This persistent scene is initialized from an input image and incorporates information from multiple generated novel view videos through an iterative process. The process combines novel view synthesis, scene integration, and scene optimization, enabling progressive expansion of the viewable area while maintaining multi-view consistency. Fig. 2 provides an overview of the overall FlexWorld framework. FlexWorld is particularly effective for extreme viewpoint changes, such as full 360° scene generation. This section details the overall framework, while additional implementation specifics are provided in Appendix B.

**Novel view synthesis.** To generate new 3D content or previously occluded objects in a persistent scene, our expansion process initiates with novel view synthesis on the existing scene. Our approach integrates existing scene information into the generation of new content to ensure geometric consistency. Specifically, we employ a video-to-video (V2V) diffusion model (see Sec. 3.3), drawing inspiration from recent advances in [10, 11], which processes rendered video from the current scene as input and produces high-quality video from the same viewpoint as output. The V2V model enables flexible control over camera trajectories hidden within incomplete input videos to generate novel views. For example, when constructing 360° scenes, we first expand the initial view via zoom-out synthesis to establish the surrounding environment. The camera then alternates between 180° left and right rotations, progressively enriching details in each iteration, achieving 360° view scenes in 3 iterations.

**Scene integration.** New 3D content extracted from generated videos is then geometry-awarely incorporated into the persistent scene. To maintain global consistency, we propose an effective empirical approach for integrating 3D structures from sequential frames. We select $m$ keyframes from the generated video to facilitate the extraction of 3D content, i.e., point cloud. We utilize DUSt3R [50] to generate initial depth maps $\hat{D}_0, ..., \hat{D}_m$ for each of the $m$ keyframes $I_1, ..., I_m$ and a reference view $I_0$ simultaneously. For each view, we render the corresponding incomplete depth maps $D_0, ..., D_m$ from the existing scene, along with their masks $M_0, ..., M_m$. The reference view is usually well optimized, and its rendered depth $D_0$ is completely known and can be used to measure the depth scale. For each $1 \leq i \leq m$, the new adding point cloud $\mathcal{P}_i$ from view $i$ can be obtained by:

$$\tilde{D}_i = \text{Depth-align} \left( \frac{\text{Median}(D_0))}{\text{Median}(\hat{D}_0))} \cdot \hat{D}_i, D_i, M_i \right), \tag{3}$$

$$\mathcal{P}_i = \{\tilde{D}_i(u,v) E_i^{-1} K^{-1} \cdot (u,v,1)^T | M_i(u,v) = 1\}, \tag{4}$$

where $E_i$ denotes extrinsic for keyframe $i$, $K$ denotes intrinsic, and $(u,v)$ stands for the pixel coordinates of the frame, ranging from 0 to frame size. Median$(\cdot)$ represents extracting the median value from given depth map. By aligning the depth scale of the reference view, we mitigate the instability inherent in the depth estimation model. Depth-align$(\cdot)$ denotes any further depth alignment operation, implemented here via guided filtering [65] for smoother transitions. The resulting point clouds $\{\mathcal{P}_1, ..., \mathcal{P}_m\}$ are finally converted to 3DGS and merged into the persistent scene.

**Scene optimization.** The expanded scene further undergoes comprehensive optimization using all available video frames generated in the latest iteration. The scene is optimized using the loss function

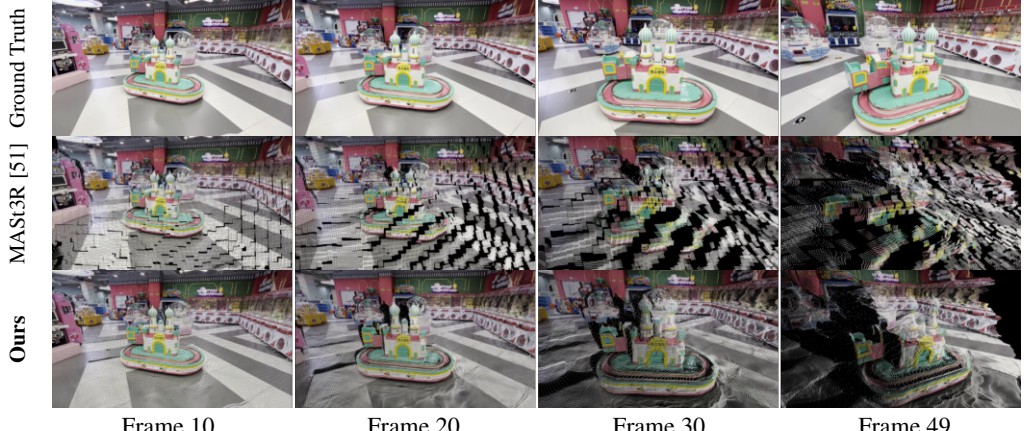

Figure 4: **Our dataset construction method yields more accurate training pairs.** We present frames of incomplete videos rendered from initial point clouds generated by a dense stereo model MASt3R [51] (i.e., ViewCrafter [10]'s dataset construction method) and our 3DGS reconstruction. Our approach produces incomplete videos with better alignment to ground truth, resulting in higher-quality training pairs.

defined in Eq. (2), with hyperparameter details provided in Sec. 4. This refinement enhances visual quality and geometric accuracy while maintaining the unified structure of the persistent representation.

In contrast to representative scene generation methods based on V2V models like ViewCrafter [10], which employs transient point cloud representations that are re-estimated and discarded per iteration, our approach maintains a persistent 3D structure throughout. This eliminates the need for repeated 3D reconstruction from images, enabling both single-image scene generation and seamless expansion of existing scenes (Fig. 1), a capability unattainable by ViewCrafter. We provide a discussion in Sec. 4.4 and compare the performance of the two frameworks in generating 3D scenes in Appendix C.2.

### 3.3 Improved diffusion for novel view synthesis

Although the framework presented in Sec. 3.2 demonstrates feasibility, we aim for the V2V model to maintain content consistency across large camera variations. This capability would enable flexible-view scene generation with fewer iterations, thereby mitigating cumulative errors that could compromise 3D scene coherence. However, existing V2V approaches [10, 11] fail to handle significant viewpoint changes (180°), as the results show in Fig. 3, primarily due to using weaker base models [57, 66] trained on suboptimal data, as shown in Fig. 4. We improve our V2V diffusion model by conducting video conditioning on an advanced base model and carefully designed training data.

**Video conditioning.** Our V2V diffusion architecture builds upon CogVideoX-5B-I2V [15], a significantly more advanced foundation than previous baselines. We modify the original architecture by replacing image conditioning with video conditioning through a 3D-VAE encoder. The encoded conditional videos are channel-wise concatenated with noise latents during diffusion. Formally, given a camera trajectory $c$, our model learns the distribution $x \sim p(x|y)$, where $y$ represents the incomplete video rendered from the coarse scene under $c$, and $x$ represents the target high-quality video. Notably, $y$ may include unobserved black regions, and we do not apply any special handling for these regions, allowing the use of the standard I2V model with minimal architectural changes. The training objective follows the standard diffusion formulation Eq. (1).

**Training data construction.** Conventional training pairs generated by dense stereo models [10, 50, 51] often contain substantial geometric inaccuracies and texture artifacts (Fig. 4), which impacted the generation quality of the trained V2V model (see Appendix A for further discussion). To overcome this limitation, we implement a synthetic data generation pipeline with improved geometric fidelity:

- Perform comprehensive 3DGS reconstruction using all available scene images;
- Select a random frame, extract its depth map via 3DGS, and back-project to obtain a point cloud;
- Render incomplete video sequences $y$ (49 consecutive frames) under continuous complex camera trajectories from datasets;

- Pair $y$ with ground truth $x$ to form training pairs $(x, y)$.

This pipeline yields superior depth estimation accuracy, producing higher-quality initial point clouds, significantly improving training pair fidelity (Fig. 4).

After training, our video model generalizes to generate high-quality novel views for coarse scenes from incomplete inputs under arbitrary trajectories, particularly under large camera variation (see Fig. 3 and Fig. 10 in Appendix E). This positions our model as the optimal video diffusion model in FlexWorld, significantly enhancing the generation of flexible-view 3D scenes.

## 4 Experiment

We present the implementation details for FlexWorld, comparison of novel view synthesis (video generation) and 3D scene generation, and show the ability of FlexWorld to perform scene extrapolation sequentially. We also perform an ablation study in Appendix C.2 and provide more results in Appendix E.

### 4.1 Implementation details

We build our video-to-video model based on the image-conditioned video diffusion model CogVideoX-5B-I2V [15]. The model is trained at a resolution of $480 \times 720$, with a learning rate of 5e-5 and a batch size of 32, for a total of 5000 steps on 16 NVIDIA A800 80G GPUs. We retain the default settings for other hyperparameters in the original I2V fine-tuning process. In the training dataset, we utilize data from the DL3DV-10K dataset [18], discarding any data with failed COLMAP camera annotations, which results in a final set of 10253 3D scenes. The coefficients for the 3DGS loss function, specifically $\lambda_1$, $\lambda_{\text{SSIM}}$, and $\lambda_{\text{LPIPS}}$, are set to 0.8, 0.2, and 0.3, respectively. More details can be found in Appendix B.

### 4.2 Comparison on novel view synthesis

We evaluate the capability of our video-to-video model for novel view synthesis by comparing the visual generation quality and camera accuracy of 5 open-source baseline models, including MotionCtrl [8], CameraCtrl [9], DimensionX [13], See3D [11], ViewCrafter [10].

**Evaluation datasets.** To ensure fairness, we selected the RealEstate10K (RE10K) test dataset [16] and Tanks-and-Temples (Tanks) [17] datasets, which are separate from our training dataset, for evaluation. Following previous work [12, 10], we randomly selected 300 video clips with a sample stride ranging from 1 to 3 in the RealEstate10K[2]. In the Tanks-and-Temples dataset, we randomly sampled 100 video clips with a stride of 4 across 14 test scenes. Notably, this dataset does not contain pre-labeled cameras; therefore, we utilized the MASt3R [51] model to annotate the cameras. Each selected video clip involves a camera length of 49. For models generating fewer than 49 frames, we uniformly excluded cameras from the original trajectory to match the required length.

**Evaluation metrics.** We followed previous works [12, 10] to evaluate the generated videos using various metrics comprehensively. The metrics include FID [67] and FVD [68] for assessing visual quality, as well as PSNR, SSIM [63], and LPIPS [64] to evaluate the similarity between the generated frames and the ground truth, with the average of the calculated metrics for each frame taken. Additionally, we estimated the corresponding camera poses for each generated frame and the ground truth using MASt3R [51] for all models. The camera accuracy was calculated using the formula from prior research [9, 10, 12].

**Qualitative comparison.** From the qualitative comparison shown in Fig. 5, all models exhibit a certain level of control over camera movements, and methods like ViewCrafter, See3D, and FlexWorld demonstrated relatively precise control; however, the visual quality of the generated outputs varied. The results from MotionCtrl often exhibited artifacts, while the content produced by CameraCtrl appeared somewhat blurred. See3D struggled to generate distinct new objects from novel viewpoints, and ViewCrafter produced dark content. In contrast, our method maintained effective camera control and surpassed all baseline models in the visual quality of the generated content.

---

[2]For V2V-based models (i.e., See3D, ViewCrafter, FlexWorld), we standardized the input by using MASt3R to estimate both the initial point cloud and the camera poses, rather than using the original poses from the dataset.

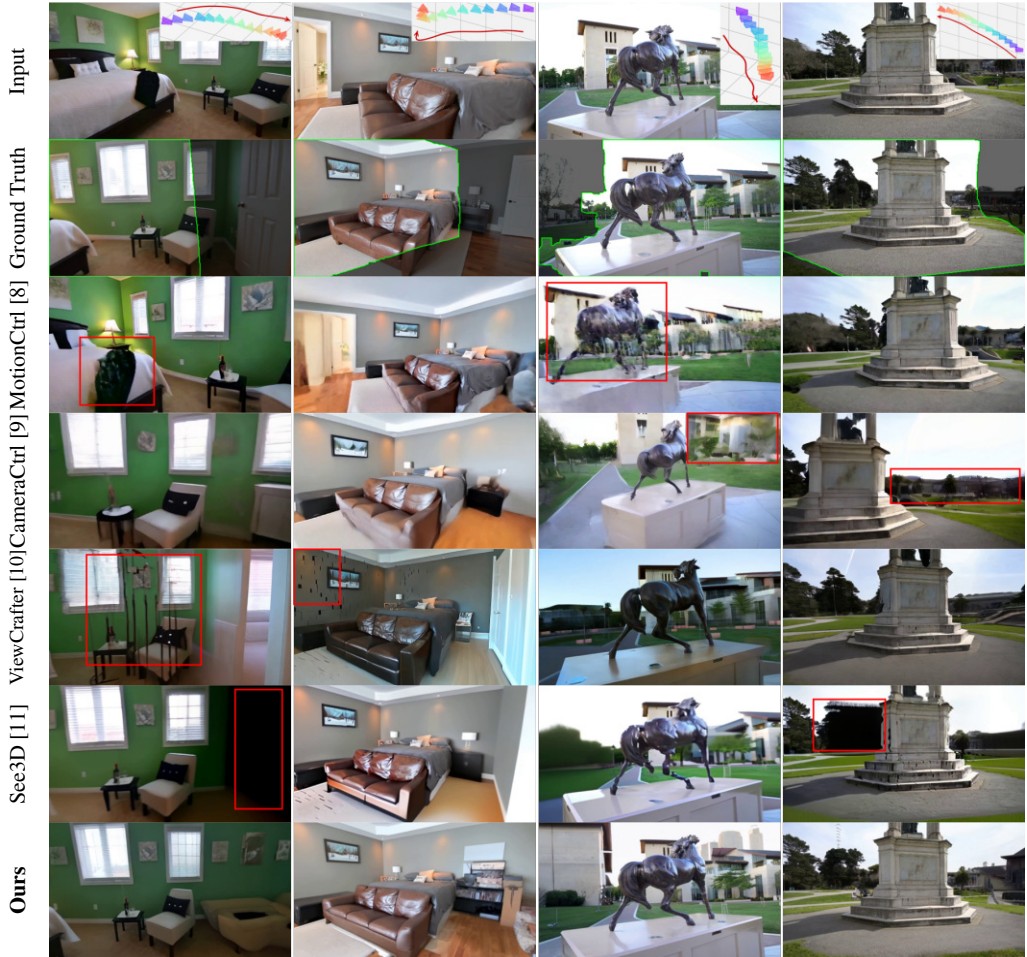

Figure 5: **Qualitative comparison on novel view synthesis.** We assessed the generated videos from various models using the same camera trajectory, focusing on the midpoint. The green bounding box in the ground truth highlights regions requiring consistency with the input, while the remaining areas demand coherent content generation. The red bounding box marks low-quality outputs in baseline models. Our model demonstrates superior visual generation quality, even under effectively controlled camera conditions.

**Quantitative comparisons.** Our quantitative results are presented in Tab. 1. FlexWorld outperforms all baselines across datasets, achieving the best FID and FVD scores, indicating that generated content distribution closely aligns with the ground truth. It also attains optimal PSNR, SSIM, and LPIPS scores, demonstrating superior visual quality. Additionally, our model excels in camera control, with lower $R_{err}$ and $T_{err}$ values.

### 4.3 Comparison on scene generation

We mainly evaluate our method for 3D scene generation by comparing the visual quality of the rendering results with 4 open-source baseline methods: LucidDreamer [4], DimensionX [13], See3D [11], and ViewCrafter [10]. Using the same sampling strategy as in Sec. 4.2, we randomly selected 100 and 50 images from the RE10K [16] and Tanks [17] datasets for evaluation. Except for LucidDreamer, which generates scenes using its original implementation, scenes for other methods are reconstructed from the videos generated to 3DGS, with reconstruction hyperparameters set in [13]. We choose PSNR, SSIM, and LPIPS for the evaluation metrics to compare the renderings from the generated 3D scenes by each baseline against the ground truth frames.

As illustrated in the qualitative comparison in Fig. 6, the scenes generated by FlexWorld exhibit higher consistency with the content of the input images compared to other baselines. Furthermore,

Table 1: **Quantitative comparison on novel view synthesis.** Our method achieves superior visual quality while maintaining commendable camera control compared to the baselines.

| Metric | FID ↓ | FVD ↓ | PSNR ↑ | SSIM ↑ | LPIPS ↓ | $R_{err}$ ↓ | $T_{err}$ ↓ |
|---|---|---|---|---|---|---|---|
| *RealEstate10K* | | | | | | | |
| MotionCtrl | 20.41 | 226.62 | 13.19 | 0.516 | 0.515 | 0.141 | 0.216 |
| CameraCtrl | 22.73 | 381.38 | 16.03 | 0.604 | 0.416 | 0.040 | **0.117** |
| DimensionX | 33.77 | 548.19 | 11.77 | 0.491 | 0.659 | 0.864 | 0.615 |
| See3D | 24.24 | 259.62 | 14.44 | 0.546 | 0.477 | **0.026** | 0.355 |
| ViewCrafter | 16.99 | 143.89 | 15.74 | 0.595 | 0.372 | 0.032 | 0.380 |
| **FlexWorld** | **13.88** | **100.41** | **16.62** | **0.612** | **0.344** | **0.026** | 0.297 |
| *Tanks and Temples* | | | | | | | |
| MotionCtrl | 54.24 | 651.47 | 11.39 | 0.361 | 0.606 | 0.336 | 0.589 |
| CameraCtrl | 60.21 | 1338.53 | 11.08 | 0.363 | 0.688 | 0.202 | 0.535 |
| DimensionX | 54.13 | 1051.15 | 11.26 | 0.358 | 0.678 | 0.878 | 0.695 |
| See3D | 53.29 | 564.19 | 12.95 | 0.404 | 0.584 | **0.035** | 0.108 |
| ViewCrafter | 41.18 | 549.10 | 12.52 | 0.386 | 0.526 | 0.111 | 0.200 |
| **FlexWorld** | **37.31** | **376.49** | **13.20** | **0.405** | **0.525** | 0.048 | **0.100** |

Table 2: **Quantitative comparison on 3D scene generation.** Scenes generated from single images by our method achieve nearly superior metric results across various datasets.

| Dataset | Results on *RealEstate10K* | | | Results on *Tanks and Temples* | | |
|---|---|---|---|---|---|---|
| Metric | PSNR ↑ | SSIM ↑ | LPIPS ↓ | PSNR ↑ | SSIM ↑ | LPIPS ↓ |
| LucidDreamer | 13.03 | 0.498 | 0.590 | 11.67 | 0.342 | 0.661 |
| DimensionX | 11.55 | 0.438 | 0.718 | 11.02 | 0.308 | 0.700 |
| See3D | 14.60 | 0.544 | 0.483 | 12.82 | **0.396** | 0.584 |
| ViewCrafter | 15.06 | 0.562 | 0.446 | 12.35 | 0.356 | 0.581 |
| **FlexWorld** | **16.18** | **0.604** | **0.369** | **12.99** | 0.389 | **0.544** |

FlexWorld generates content with higher visual quality in new regions beyond the input. We also conducted a quantitative comparison, as presented in Tab. 2, which shows that FlexWorld outperforms nearly all baselines in terms of metrics, with only a slight decrease compared to See3D on the SSIM in the Tanks [17] dataset. All results indicate that FlexWorld generates scenes with higher 3D consistency and visual quality.

### 4.4 Scene extrapolation

Leveraging the progressive expansion process, FlexWorld can extend a given 3D scene into a larger, more flexible-view one, distinguishing it from methods like ViewCrafter [10]. FlexWorld can start with a 3DGS scene and iteratively expand it, as shown in Fig. 1c with 3D scenes reconstructed from DL3DV[18], which FlexWorld further extrapolates into larger scenes. Our approach can also resolve artifacts such as holes or blurriness in the original, highlighting its scalability in generating high-quality scene extrapolation. See Appendix B.3 for detailed implementation and Fig. 13 in Appendix E for more results.

### 4.5 Analysis of flexible camera control

FlexWorld is capable of generating novel view videos under flexible camera control. As quantified in Tab. 1, it excels on challenging real-world datasets like RE10K [16] and Tanks [17]. This advantage is particularly evident on the highly varied, complex, and non-linear camera paths characteristic of these benchmarks. Furthermore, we explicitly demonstrate its robustness to large camera motions. As shown in Fig. 3 and Appendix Fig. 10, our model maintains high-fidelity synthesis even under significant camera variations. Finally, the extensive result gallery in Appendix Fig. 12 provides additional qualitative evidence, highlighting the model's versatility across a wide array of distinct camera trajectories.

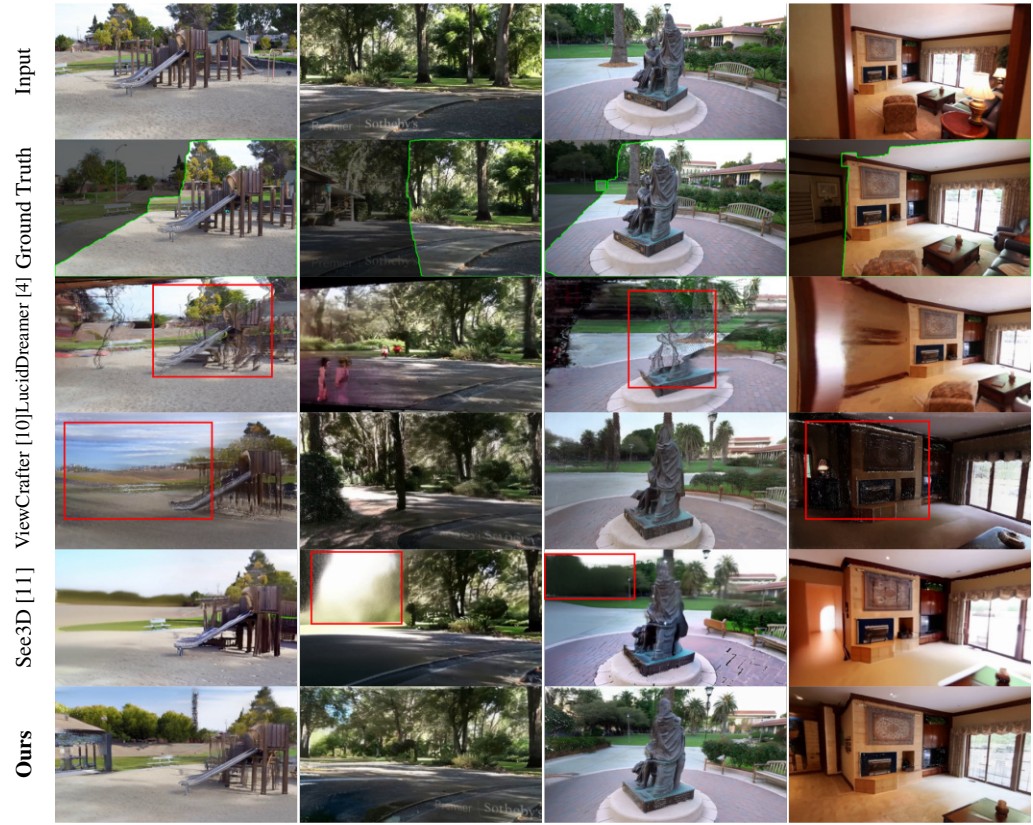

Figure 6: **Qualitative comparison on 3D scene generation.** We present images rendered from scenes generated by various single image-to-3D methods. The green and red bounding boxes have the same meaning as in Fig. 5. Our approach achieves superior visual results.

## 5 Conclusion

We propose FlexWorld, a framework for flexible-view 3D scene generation from single images using a persistent 3D Gaussian representation. Our approach progressively expands this representation through novel view synthesis with a fine-tuned V2V diffusion model, enabling robust handling of large viewpoint changes while maintaining visual consistency. Extensive experiments show FlexWorld's superior viewpoint flexibility and visual quality performance compared to baselines.

**Limitations and broader impact.** While FlexWorld shows promise for flexible 3D generation, limitations remain. The V2V model may lose camera control when the input lacks 3D information, though this can be alleviated by trajectory design. Additionally, FlexWorld's 3D consistency is affected by the dense stereo model's accuracy. Meanwhile, the long generation time of the video diffusion model and the lengthy optimization process of iterative 3D Gaussian splatting constrain the efficiency of generating a single scene. Nevertheless, we believe that FlexWorld is promising and holds significant potential for VR and 3D tourism. As a generative method, our method may be misused for data fabrication, necessitating strong safeguards against misuse.

## 6 Acknowledgements

This work was supported by the Beijing Natural Science Foundation (No. L247030); the National Natural Science Foundation of China (Nos. 92470118, 62306163); the Beijing Nova Program (No. 20230484416); the ByteDance Research Fund; the Public Computing Cloud of Renmin University of China; the Beijing Major Science and Technology Project under Contract no. Z251100008425002; and the fund for building world-class universities (disciplines) of Renmin University of China.

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

## A   Impact of training data in video diffusion

Our data generation methodology is guided by the core principle of maintaining consistency between training and inference conditions. During inference, our model is designed to process incomplete views rendered from an optimized 3DGS scene with well-posed geometry. Therefore, directly using training pairs from conventional dense stereo models [10, 50, 51] would introduce a significant domain gap. As shown in Fig. 4, these methods often produce training data with notable geometric inaccuracies and texture artifacts. More critically, these imperfections may be learned by the model and propagate into the final outputs, as evidenced by the artifacts in Fig. 7 which mirror those in the training data.

To further validate our approach, we conducted a brief experiment. We trained two models on the first 1K samples of the DL3DV [18] dataset, one using training pairs generated by MASt3R [51] and the other by our methodology. Despite the limited training (1000 steps with a batch size of 8 on 8 A800 GPUs, compared to the full training detailed in Sec. B.1), the visual results in Fig. 8 clearly show that the model trained with our data generates qualitatively superior 360° 3D scenes. This confirms that by aligning the training data with the inference conditions, our data generation methodology better harmonizes with the FlexWorld framework, ultimately enhancing the quality of the generated outputs.

## B   More implementation details

We present the detailed implementation details for FlexWorld in this section, including V2V model training, the complete Flexible-view 3D scene generation workflow, the process of scene extrapolation, and the codebase and safeguards.

### B.1   Training of V2V model.

The V2V model in FlexWorld builds upon CogVideoX-5B-I2V [15] and is fine-tuned using the SAT framework. Unlike the original I2V model, which processes a single image encoded by a 3D-VAE into latents with a temporal dimension of 1 (later padded with zero tensors to match compressed video dimensions), our V2V model directly accepts video input. This eliminates the need for zero-padding, as the 3D-VAE naturally encodes the temporal dimension of the input video into compressed latents. As described in Sec. 4.1, we train the video-to-video model at a resolution of $480 \times 720$, with a learning rate of 5e-5 and a batch size of 32, for a total of 5000 steps on 16 NVIDIA A800 80G GPUs, requiring approximately 70 hours to complete.

As for the training datasets, to support the generation of static scenes and large camera variations, we select the high-quality DL3DV-10K [18] scene dataset, which contains various camera movements. After a rigorous filtering process that excluded scenes with low image resolution (below $540 \times 960$), missing camera parameters, or significant pose discrepancies between the official DL3DV annotations and a COLMAP reconstruction cache, we obtained 10253 high-quality 3D scenes for training. Each scene was reconstructed into 3D Gaussians using the official implementation, optimized for 7000 steps (taking about 5 minutes per scene on an NVIDIA A800 GPU). The full dataset reconstruction was completed in roughly 4-5 days by parallelizing across 8 A800 GPUs. We exclude the RealEstate10K dataset [16] from our training dataset, as its videos frequently contain moving objects and simple camera motions, which fail to meet our needs.

### B.2   Flexible-view 3D scene generation

FlexWorld begins with constructing an initial 3D point cloud from a single input image using DUSt3R [50]. Since dense stereo requires paired images, we duplicate the single input image to serve as both the source and reference views.

As for scene representation, we employ 3DGS [20] as the core representation, utilizing gsplat [69] for implementation. This choice is motivated by two fundamental advantages over simpler point-cloud representations: rendering quality and functional capability. While a raw point cloud serves as a geometric scaffold, rendering it directly often produces sparse results with holes and lacks photorealism. In contrast, 3DGS models not only geometry but also local appearance, enabling the synthesis of high-fidelity, anti-aliased novel views.

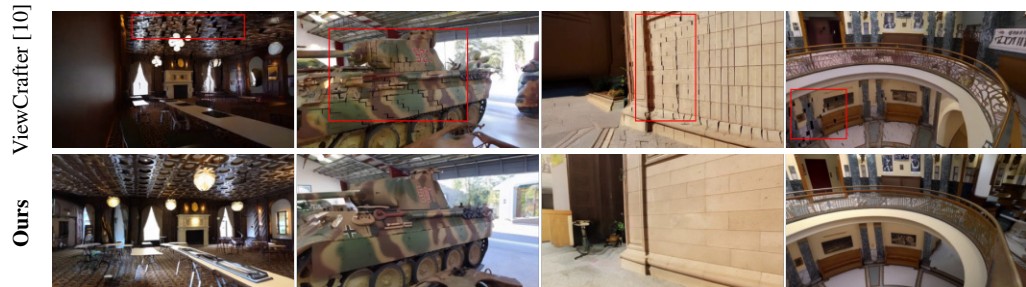

Figure 7: **Artifacts generated by ViewCrafter [10].** Compared to FlexWorld, ViewCrafter produces more artifacts that resemble those found in the incomplete videos within the training dataset constructed by its method.

When processing point cloud data (e.g., the initial point cloud), it will be immediately converted into 3DGS, serving as the initial scene representation. Unlike the original 3DGS, which uses spherical harmonics, our implementation directly represents color using RGB values. We avoid downsampling during the initialization of 3DGS from the point cloud, so the number of Gaussian counts equals the number of point clouds. Gaussian properties are initialized directly from the point cloud's position and color, with scale and opacity set to isotropic values of 3e-4 and 0.8, respectively, and rotation initialized using the identity matrix.

For novel view synthesis, camera trajectories are interpolated between the first and last frames to produce 49 poses, matching the V2V model's input requirements. Spatial coordinates use linear and cubic spline interpolation, while rotation matrices employ spherical interpolation for smooth transitions. To avoid collisions, the movement range is constrained by the minimum depth derived from the input image's depth estimation.

When integrating new 3DGS into an existing scene, we utilize DUSt3R [50] to extract consistent depth from keyframes. We select $m = 6$ keyframes and employ the fully connected pairing strategy to achieve more accurate depth estimation. Keyframes are selected deterministically using a uniform sampling strategy, with the reference view typically chosen as the input image's corresponding view due to its superior visual quality and role as the starting point for scene expansion. After depth alignment, we utilize alpha maps as masks rendered from the scene to avoid the inclusion of redundant content. We apply 25 iterations of dilation to the alpha map to mitigate fragmentation in the added points.

During 3DGS optimization, we enhance visual quality by upscaling input video frames using the image super-resolution model Real-ESRGAN [70]. We use the original Gaussian paper's strategies for splitting, duplicating, and pruning Gaussians, but we disable the reset opacity strategy. Compared to the original Gaussian, we use higher learning rates: 1e-5 for position, 5e-3 for color, 5e-2 for opacity, 5e-4 for scale, and 1e-4 for rotation.

To further enhance the visual quality of the generated scene, we adopt SDEdit [71] by rendering multi-view images $I$ from fixed viewpoints, adding random noise, and applying a multi-step denoising process using the FLUX.1-dev [72] image diffusion model after the expansion of the scene. During the refinement process, the timestamp for the forward diffusion process is set at $0.6T$, where $T$ represents the total duration of the diffusion process. We focus on refining images when rotating cameras rather than translating them. Specifically, we refine 5 frames from a panoramic scene using image-to-image refinement, followed by 1000 iterations of optimization with the same loss function in Eq. (2) across all images to refine the overall Gaussian representation.

FlexWorld's full workflow is an iterative process that maintains a persistent 3DGS scene as a coherent anchor. This ensures newly generated content aligns with the established structure, effectively reducing accumulated errors common in methods lacking such geometric memory. As described in Sec. 3.2, the complete FlexWorld pipeline requires three iterations to generate a 360° scene. When executed on a single NVIDIA A800 GPU, the entire process takes approximately 30 minutes.

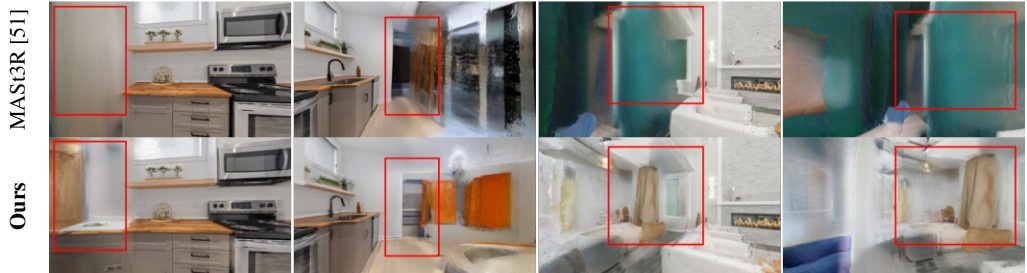

Figure 8: **Comparison of 360° scene generation results with different data generation methodologies.** The methodology proposed by FlexWorld yields more structurally consistent and coherent content compared to dense stereo models, such as MASt3R [51].

Table 3: **Codebase.** We provide the URL and licenses for the open-source assets we used.

| Asset | URL | License |
|-------|-----|---------|
| **Models used in FlexWorld** | | |
| [15] | `https://github.com/THUDM/CogVideo` | Apache-2.0 license |
| [50] | `https://github.com/naver/dust3r` | CC BY-NC-SA 4.0 license |
| [51] | `https://github.com/naver/mast3r` | CC BY-NC-SA 4.0 license |
| [69] | `https://github.com/nerfstudio-project/gsplat` | Apache-2.0 license |
| [72] | `https://github.com/black-forest-labs/flux` | Apache-2.0 license |
| [70] | `https://github.com/xinntao/Real-ESRGAN` | BSD-3-Clause license |
| **Baselines** | | |
| [8] | `https://github.com/TencentARC/MotionCtrl` | Apache-2.0 license |
| [9] | `https://github.com/hehao13/CameraCtrl` | Apache-2.0 license |
| [13] | `https://github.com/wenqsun/DimensionX` | Apache-2.0 license |
| [11] | `https://github.com/baaivision/See3D` | Apache-2.0 license |
| [10] | `https://github.com/Drexubery/ViewCrafter` | Apache-2.0 license |
| [4] | `https://github.com/luciddreamer-cvlab/LucidDreamer` | CC BY-NC-SA 4.0 license |
| **Datasets** | | |
| [18] | `https://github.com/DL3DV-10K/Dataset` | CC BY-NC 4.0 license |
| [16] | `https://google.github.io/realestate10k` | CC BY 4.0 license |
| [17] | `https://www.tanksandtemples.org` | CC BY 4.0 license |

## B.3 Scene extrapolation

Given an existing 3DGS scene, FlexWorld can be directly applied to expand it into a larger, more flexible-view one. We directly use the input 3DGS scene as the initial persistent representation, rather than producing it with a single image. Then, through multiple iterative steps as described in Section 3.2, we progressively expand the scene, ultimately generating a larger scene.

To evaluate the capability of FlexWorld, we use the scene reconstructed from DL3DV as the input. The extrapolation uses two iterations with camera trajectories alternating between 180° left and right rotations, and the process takes approximately 20 minutes on a single NVIDIA A800 GPU.

## B.4 Codebase and safeguards

Table 3 lists the URLs and licenses of the open-source resources used in this paper, including models used in FlexWorld, baselines for comparison, and training/evaluation datasets.

Notably, our work involves fine-tuning the publicly available CogVideoX [15] model on the DL3DV [18] dataset, which contains almost no unsafe images. To ensure responsible usage, we will implement safeguards by enforcing strict controlled-use requirements when publicly releasing these resources.

Table 4: **Quantitative comparison of novel view video generation using the WorldScore benchmark.** For all metrics, higher scores indicate better performance.

| Methods | WorldScore (Overall) | Camera Ctrl | Object Ctrl | Content Align | 3D Consist | Photo Consist | Style Consist | Subjective Qual |
|---|---|---|---|---|---|---|---|---|
| ViewCrafter | 66.81 | 78.25 | 48.92 | 50.63 | 81.88 | 74.90 | 68.05 | 65.01 |
| See3D | 62.86 | 85.21 | 55.40 | 54.46 | 86.22 | 85.05 | 44.74 | 28.91 |
| **FlexWorld** | **68.37** | 82.79 | 44.92 | 49.35 | 84.70 | 90.53 | 64.36 | 61.97 |

Table 5: **Quantitative comparison of 3D scene generation using the WorldScore benchmark.** Similar to the video evaluation, higher scores are better across all metrics.

| Methods | WorldScore (Overall) | Camera Ctrl | Object Ctrl | Content Align | 3D Consist | Photo Consist | Style Consist | Subjective Qual |
|---|---|---|---|---|---|---|---|---|
| ViewCrafter | 66.20 | 81.02 | 69.00 | 47.77 | 83.46 | 81.30 | 64.88 | 35.94 |
| See3D | 59.48 | 81.22 | 57.42 | 44.94 | 82.97 | 83.15 | 46.38 | 20.30 |
| **FlexWorld** | **67.65** | 80.35 | 69.50 | 46.27 | 87.47 | 88.42 | 61.72 | 39.84 |

# C   Additional experimental results

In this section, we present additional experiments to assess our model's performance further, followed by a detailed ablation study to validate our design choices.

## C.1   Evaluation with WorldScore

To provide a more comprehensive assessment of generation quality, we conduct further evaluations using a 3D-centric metric WorldScore benchmark [73], with the two main baselines, See3D [11] and ViewCrafter [10]. For the video generation task, we test on a diverse subset of 300 scenes, created by selecting the first 15 scenes from each of the 20 available categories. This ensures a representative sample covering various conditions (e.g., indoor/outdoor, stylized/photorealistic). For the more computationally intensive 3D scene generation task, we evaluate on the first 100 scenes from this subset. All evaluations strictly adhere to the official WorldScore protocol, and the generation methods for all models are consistent with those described in Sec. 4.2 and 4.3.

The results are presented in Tab. 4 and 5. Our model achieves a superior overall WorldScore in both video and 3D scene generation, underscoring its robust performance. It is noteworthy that our method demonstrates significant advantages even on the WorldScore benchmark, which features relatively simple camera trajectories. This highlights the general effectiveness and versatility of our approach beyond complex camera paths.

## C.2   Ablation study

We conduct comprehensive ablation studies to validate the effectiveness of each key component in FlexWorld. We provide both qualitative results in Fig. 9 and a quantitative analysis in Tab. 6. For the quantitative study, we again utilize the WorldScore benchmark [73] to evaluate videos rendered from the generated 360° scenes. Due to computational constraints, this analysis was performed on a subset of 155 test cases (the first 31 images from the first 5 categories). We evaluate five main configurations to isolate the contribution of each component.

**Ablation on video diffusion.** As shown in Fig. 9a, replacing our V2V model in FlexWorld with ViewCrafter resulted in blurred scene content. This is due to inconsistencies in ViewCrafter's output under large camera variations, as discussed in Sec. 3.3. The quantitative results in Tab. 6 (row 1 vs. row 4) corroborate this finding, showing a substantial drop in performance across all metrics.

**Ablation on camera trajectory.** A zoom-out movement is crucial for enlarging the scene to enhance camera control. Without it, the generated video will mismatch with the input trajectory, leading to inconsistencies and blurriness in the generated scene, as shown in Fig. 9b.

Table 6: **Quantitative ablation study using the WorldScore benchmark.** We analyze the impact of different components and framework configurations. Higher is better for all metrics, and the Camera Control metric is not applicable as our final scene camera movements are more complex than the WorldScore defaults.

| Configuration | WorldScore (Overall) | Object Ctrl | Content Align | 3D Consist | Photo Consist | Style Consist | Subjective Qual |
|---|---|---|---|---|---|---|---|
| Viewcrafter's V2V + our framework (w/o refine) | 44.12 | 65.81 | 39.93 | 61.40 | 0.00 | 89.40 | 52.33 |
| Our V2V+ ViewCrafter style's framework (w/o refine) | 49.97 | 70.16 | 55.66 | 62.66 | 37.77 | 86.60 | 36.95 |
| Our framework (w/o super-resolution, w/o refine) | 56.55 | 72.74 | 57.25 | 69.85 | 51.72 | 96.30 | 47.96 |
| Our framework (w/o refine) | 56.34 | 73.06 | 58.30 | 71.07 | 46.62 | 96.48 | 48.84 |
| Our full framework (w/ refine) | 55.60 | 71.29 | 57.09 | 70.81 | 45.69 | 96.66 | 47.67 |

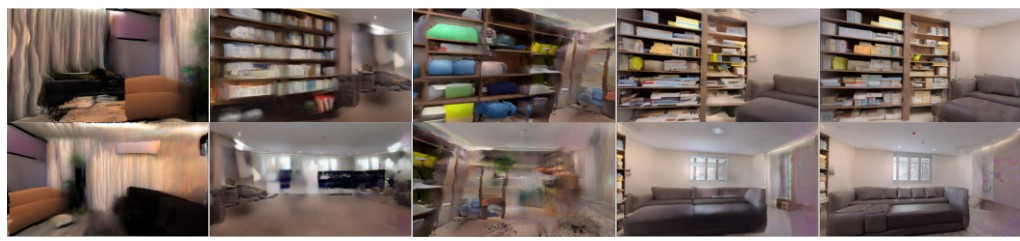

(a) w/o V2V    (b) w/o zoom-out    (c) w/o framework    (d) w/o refine    (e) Full

Figure 9: **Ablation study.** To generate a 360° view 3D scene, FlexWorld necessitates our video-to-video model, an initial zoom-out trajectory and generation framework. Additionally, a refinement process can further enhance the visual quality of the generated 3D scene.

**Ablation on generation framework.** As the scene generation framework of ViewCrafter [10] remains unopen, we reimplement its framework following its described pipeline and generate 3D scenes on our V2V model for fair comparison. When using our V2V model with Viewcrafter's framework, the resulting scene quality is visibly lower (Fig. 9c) and scores poorly in the quantitative evaluation (Tab. 6, row 2 vs. row 4). In contrast, our proposed framework not only produces higher-quality results but also uniquely supports scene extrapolation, as detailed in Section 3.2.

**Ablation on super-resolution.** We optionally incorporate a Real-ESRGAN super-resolution module to enhance the texture detail of the generated videos. However, our ablation in Tab. 6 (comparing rows 3 and 4) reveals that this step has a negligible impact on the WorldScore metrics. This suggests that while it may improve final visual appeal, it is not essential for constructing the core 3D geometry and structure.

**Ablation on refinement process.** The video model's generation quality restricts the detail in the generated scene. A refinement process, as detailed in Appendix B, further modestly enhances the generated visual details while preserving the existing geometric structure of the scene, as shown in Fig. 9d. And the process slightly influences metrics in WorldScore (comparing rows 4 and 5 in Tab. 6).

# D   Limitations and future work

A limitation of our current framework is its computation time of approximately 30 minutes per scene (see Appendix B.2). Notably, this is comparable with previous work (e.g., ViewCrafter) under similar conditions. Our observations indicate that the primary bottlenecks are the video diffusion model's generation time and the iterative 3D Gaussian splatting optimization. Furthermore, we recognize that inaccuracies in the initial depth estimation may introduce accumulated errors during the reconstruction process. Crucially, these limitations can be substantially mitigated by leveraging rapid advancements in foundational models. We are confident that this limitation will be alleviated by leveraging advances in base models, such as distilled video generators [74] for accelerated sampling and feed-forward reconstruction 3DGS models [75] to bypass iterative optimization. Similarly, the impact of accumulated errors could be reduced by employing improved depth estimators (e.g., VGGT [76]). Integrating these is a promising direction for future work.

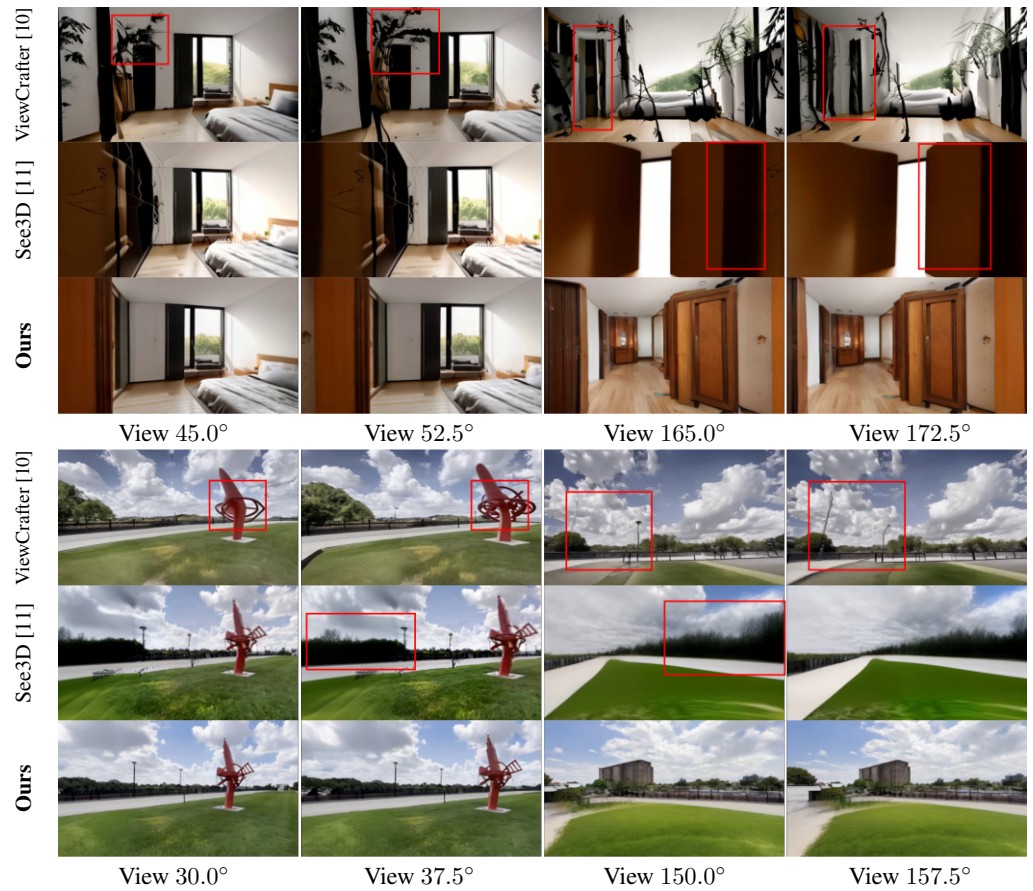

Figure 10: More comparative results showcasing generated videos under large camera variations.

# E More results

We present more results of comparison on generated videos under large camera variations in Fig. 10, our video generation in Fig. 12, the 360° 3D scene generation in Fig. 11, and the 3D scene extrapolation in Fig. 13.

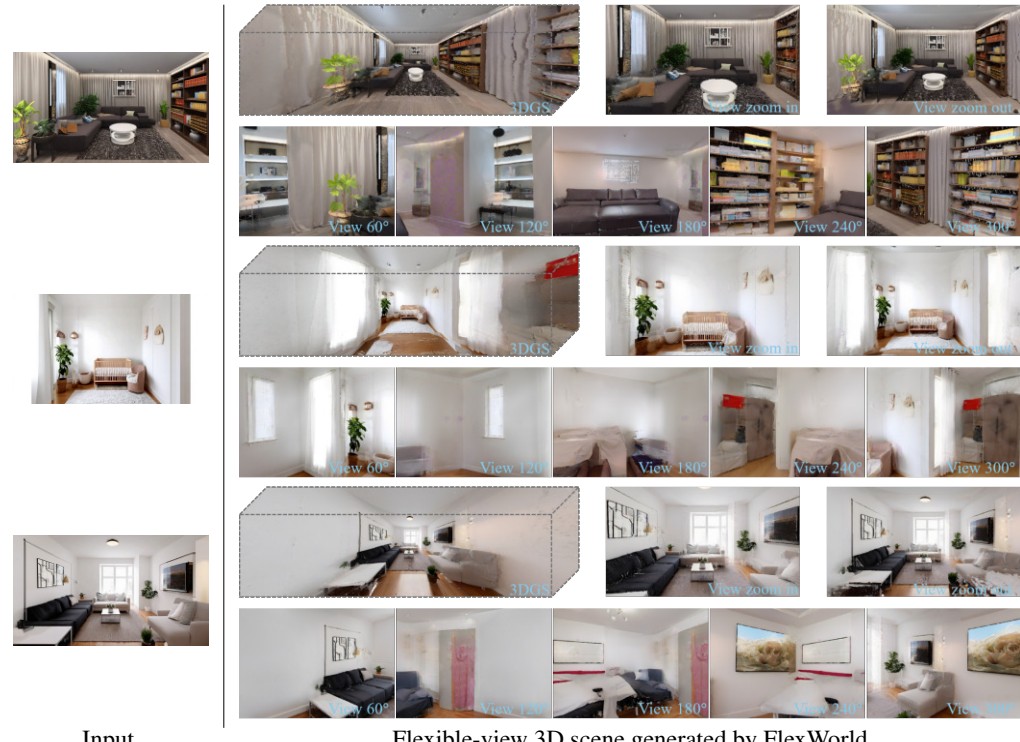

Input           Flexible-view 3D scene generated by FlexWorld

Figure 11: More results of generated 360° scene from FlexWorld.

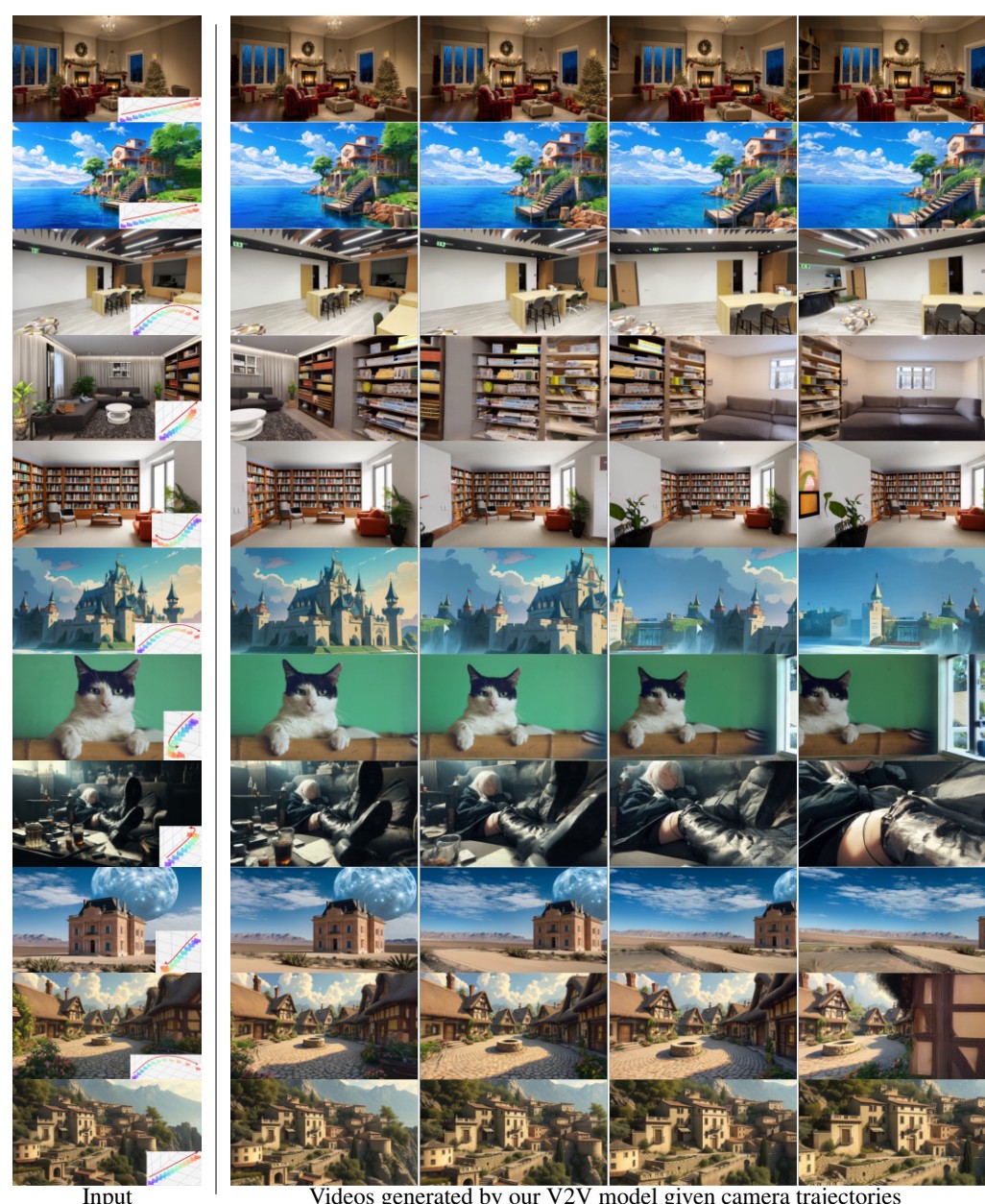

Input    Videos generated by our V2V model given camera trajectories

Figure 12: More results of generated videos from FlexWorld.

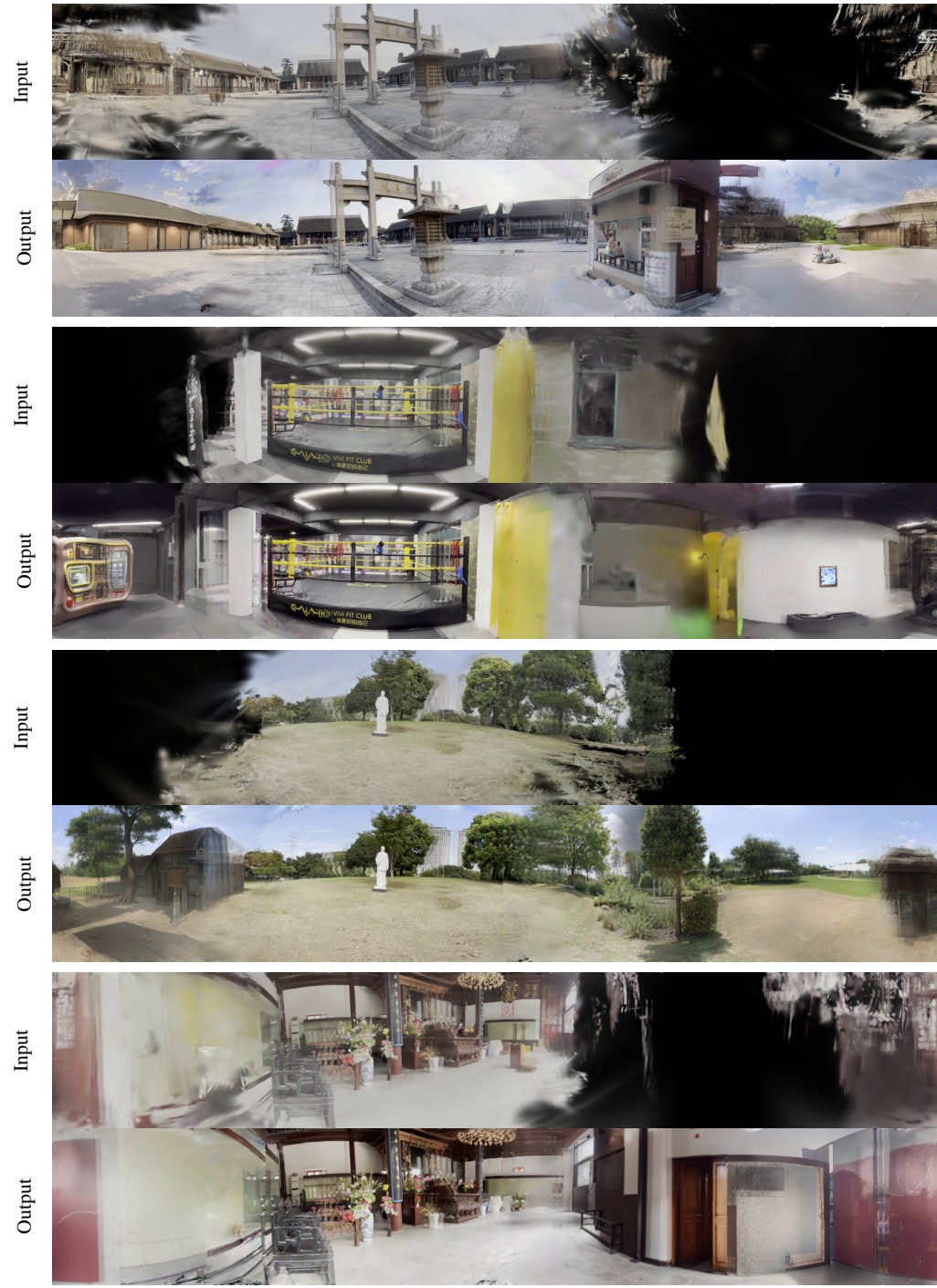

Figure 13: **Scene extrapolation.** We show FlexWorld's ability to extend existing scenes beyond their original boundaries. The results are presented in 360° panoramas, where the top image in each line illustrates the incomplete original scene, and the bottom image reveals the extrapolated one generated by FlexWorld.

