# OpenReview forum: "FlexWorld: Progressively Expanding 3D Scenes for Flexible-View Exploration"
_NeurIPS.cc/2025/Conference — NeurIPS 2025 poster_

### Official Review · Reviewer_gXbR · 2025-06-07

**Clarity:** 2
**Significance:** 3
**Originality:** 2
**Rating:** 4
**Confidence:** 4

**Summary:**

This paper presents a pipeline for generating a complete 3D scene from a single image. The core component is a video diffusion model trained on a carefully curated dataset designed to ensure view consistency. Based on this model, the authors employ 3D Gaussian Splatting (3DGS) to progressively fuse information from the diffusion model in a geometry-consistent manner.

**Questions:**

* Ln 122 reads like the proposed V2V model takes camera trajectory as input, but from Ln 164, camera trajectory c is only used for rendering videos with missing geometry y, is this correct?
* Is the camera trajectory mentioned in Line 124 the only trajectory used across all experiments?
* Is Figure 4 the output of the proposed V2V model or from the optimized 3DGS? I find the term “novel view synthesis” confusing, maybe specify if the video is rendered from 3DGS or generated with the diffusion model.
* How is y obtained in Line 173? Is it rendered from the optimized 3DGS? If so, how do you ensure that y contains missing information or artifacts? My understanding is that y and x share the same camera trajectory from the original dataset, and therefore y should be very similar to x.

**Ethical Concerns:**

["NO or VERY MINOR ethics concerns only"]

**Final Justification:**

My main concerns were that some of the key claims, such as support for flexible camera control, were not well-supported by experiments, and the dataset generation process was not clearly explained or sufficiently discussed.The authors have agreed to polish the paper further, emphasizing the data generation pipeline and including experiments to better support the claim of flexible camera support. Therefore, I am updating my rating from borderline reject to borderline accept. My remaining concern is that the method relies heavily on the 3DGS representation and a V2V model, both of which are based on prior work. In my view, the main novel contributions lie in the proposed data generation process and the way generated images are merged into the 3DGS representation, both of which may have limited novelty.

**Limitations:**

yes

**Quality:**

3

**Strengths And Weaknesses:**

Strengths:
* The proposed dataset construction pipeline helps improve the quality of training data and subsequently enhances the performance of the video diffusion model.
* Qualitative results show that the presented diffusion model achieves better view consistency compared to prior work.

Concerns:
* While the results look promising, I find some parts of the writing unclear and a bit hard to follow. In particular, many sentences are overly long and could be made more concise. Additionally, reiterating the problem motivation in the Method section disrupts the flow and makes the technical content harder to follow, for example, Ln 117–119 and 127–129 could be more compact. At the beginning of Section 3.2, figure 2 can be used to help explain the overall pipeline. Currently figure 2 is not referenced in the text. Additional clarity-related questions are listed below.
* The authors claim that the proposed method allows flexible camera control, however, this is not clearly supported by the experiments, as there is no analysis on the effect of different camera trajectories.
* It appears that the improvement in the video diffusion model primarily stems from the proposed dataset generation pipeline. Since this is one of the main contributions of the paper, it would be helpful to include more in-depth discussion and experimental analysis of this dataset generation pipeline in the main paper.

---

> ### Author Rebuttal · Authors · 2025-07-31
>
> We sincerely thank you for the thorough, insightful, and highly constructive review. Your detailed feedback has been incredibly helpful in strengthening our manuscript. We have carefully considered all your comments and questions, and we provide our point-by-point responses below.
>
> #### **Weakness 1: Writing Clarity**
>
> We sincerely thank you for your valuable and detailed feedback on the manuscript's writing and structure. We agree that clarity and conciseness are paramount for conveying our work effectively. Based on your constructive suggestions, we will undertake a thorough revision of the manuscript.
>
> Specifically, we will perform a full review to shorten overly long sentences and improve overall readability. The specific sentences you highlighted will be revised. For example, the sentence on Lines 117–119 will be revised from "Unlike most methods that generate solely from given images..., our approach integrates current scene information into newly generated content." to a more direct statement, such as: **"Our approach integrates existing scene information into the generation of new content to ensure geometric consistency."** Similarly, the sentence on Lines 127–129 will be revised from "While deriving 3D structures from sequential frames and maintaining consistency... presents significant challenges..., we propose an effective empirical approach." to focus on our solution, for example: **"To maintain global consistency, we propose an effective empirical approach for integrating 3D structures from sequential frames."**
>
> Furthermore, as you correctly pointed out, we will explicitly reference **Figure 2** at the beginning of Section 3.2 to provide readers with a clear, high-level overview of our framework from the outset. We are grateful for these actionable comments and are confident these revisions will substantially improve the clarity of our paper.
>
> #### **Weakness 2: Support for Flexible Camera Control**
>
> We thank you for this valid and constructive criticism. We agree that our initial manuscript claimed "flexible camera control" without providing a dedicated analysis to substantiate it. Your feedback has helped us significantly improve the validation of this key strength of our method.
>
> To rectify this, we will add a new subsection to the paper (e.g., Section 4.5) titled "**Analysis of Flexible Camera Control**." In this new section, we will consolidate and reframe existing evidence from our paper to explicitly support our claim:
>
> * **Performance on Diverse and Complex Trajectories**: We first argue that our model's superior performance on challenging real-world datasets like RealEstate10K and Tanks and Temples is strong evidence of its flexibility. These datasets usually feature highly varied, complex, and non-linear camera paths. Our quantitative results on these datasets, presented in Table 1, demonstrate that our model robustly handles these diverse trajectories compared to other baselines.
> * **Robustness to Large Camera Variation**: We now explicitly draw the reader's attention to figures that specifically test the limits of camera motion. We highlight that Figure 3 and Appendix Figure 9 showcase our model's ability to maintain high quality even under large camera variations, a key aspect of flexible control.
> * **Qualitative Showcase of Trajectory Diversity**: Finally, we point to the gallery of results in Appendix Figure 11, which provides extensive qualitative evidence of our model's performance across a wide array of different camera paths, further underscoring its versatility.
>
> By consolidating these results into a focused analysis, we believe we have now provided clear and compelling support for our claim of flexible camera control. We thank the reviewer again for prompting this important addition to our paper.
>
> #### **Weakness 3: Discussion of Dataset Generation Pipeline**
>
> We thank the reviewer for this constructive suggestion. We agree that our data generation process is a crucial aspect of our work that warrants a more detailed discussion. Our data generation methodology is guided by the principle of **maintaining consistency between training and inference conditions**. During inference, our model is designed to handle incomplete views that are rendered from an optimized 3DGS scene with well-posed geometry. Adopting a simpler approach, such as directly generating training data from a single-shot depth estimator, would introduce significant artifacts (as shown in Figure 6 of our paper), creating a domain gap between the training data and the inference conditions. To avoid this discrepancy, our pipeline first creates a high-quality 3DGS scene from real views. From this clean 3DGS, we then extract accurate depth maps, which are back-projected to form incomplete 3D scenes. Finally, we render "incomplete" views from the scene. This process ensures that our training data closely aligns with the conditions our model will encounter during inference.
>
> Nevertheless, we wish to emphasize the importance of the other components in our framework. As quantitatively demonstrated in our newly added quantitative ablation study (see response to reviewer v55n), our trained V2V model and our unique framework design each contribute significantly and independently to the final performance. For instance, the comparisons between configurations **(1)** & **(4)** and **(2)** & **(4)** in the table isolate and confirm the substantial impact of our V2V model and framework, respectively.
>
> We also acknowledge the valuable suggestion regarding the experimental analysis of the dataset generation. While we are keen to conduct this analysis, the rebuttal period does not provide sufficient time to construct a new training dataset using a single-shot depth estimator like MASt3R and complete the subsequent V2V model training. Therefore, we are currently unable to include this specific experiment in the ablation study. We sincerely apologize for this omission and commit to incorporating a thorough analysis of this aspect in the final version of the manuscript.
>
>
> #### **Question 1: Camera Trajectory as Input**
>
> Thank you for your sharp observation and for pointing out this confusing statement. We sincerely apologize for the lack of clarity.
>
> The camera trajectory `c` is used exclusively to render the incomplete video `y` (as described in Ln 164), which then serves as the input for the V2V model. The V2V model itself does not take `c` as a separate, direct input.
>
> The statement on Ln 122 was poorly phrased and was the source of this misunderstanding. We intended to convey that the control over the camera path is implicit in the input video, rather than an explicit condition.
>
> We will correct this in the revised manuscript. The sentence on Ln 122 now reads: **"The V2V model enables flexible control over camera trajectories hidden within input incomplete videos to generate novel views."**
>
> #### **Question 2: Only One Camera Trajectory?**
>
> Thank you for your question, which helps us clarify the flexibility of our method. We apologize if the wording on Line 124 caused any misunderstanding.
>
> The trajectory mentioned in Line 124 is not the only one used across all experiments. The trajectory described on Line 124 is a specific example for generating 360-degree scenes.
>
> Our method is designed to be general and is not constrained to any single type of camera path. As shown in other experiments, particularly in Figure 4 and Figure 5, our model successfully handles more diverse and arbitrary camera trajectories to complete generation tasks.
>
> #### **Question 3: Figure 4 Output and "Novel View Synthesis"**
>
> Thank you for pointing out this ambiguity regarding “novel view synthesis”. We sincerely apologize for the confusion. Figure 4 shows the direct output of our proposed V2V (video diffusion) model, not the views rendered from the optimized 3DGS scene. Our use of the term "novel view synthesis" was intended to be consistent with prior work, specifically ViewCrafter [1], where it describes the task of generating new views as a video sequence.
>
> To resolve this ambiguity, we will revise the caption of Figure 4 to state that the results are videos generated by our video diffusion model to avoid misunderstanding.
>
> [1] Yu W, Xing J, Yuan L, et al. Viewcrafter: Taming video diffusion models for high-fidelity novel view synthesis[J]. arXiv preprint arXiv:2409.02048, 2024.
>
> #### **Question 4: How the Incomplete Video `y` is Obtained**
>
> Thank you for your question. To clarify, your understanding that `y` and `x` share the same camera trajectory is correct. However, the 3D structure used to render `y` is built using only the information from the first frame of the video, as stated on Ln 172. We create a sparse point cloud from this single frame, which naturally lacks information about parts of the scene outside the initial view. The "incomplete video" `y` is the result of rendering this sparse structure along the full camera path.
>
> Consequently, `y` is guaranteed to contain significant missing regions and artifacts compared to the complete ground truth video `x`, making it a suitable input for our V2V task.
>
>
> We sincerely appreciate your time and insightful feedback. Your comments have been invaluable in improving our manuscript. We have carefully addressed all your concerns through additional analyses and clarifications, which we believe will strengthen the paper. While some suggested experiments will require more time than is available for this rebuttal, we are confident the current modifications have substantially improved our work.

---

> > ### Comment · Reviewer_gXbR · 2025-08-03
> >
> > Thank you for addressing all my concerns, I do not have further questions. I will take the rebuttal into account as I finalize my review.

---

> > > ### Author Response · Authors · 2025-08-04
> > >
> > > Thank you for your time and constructive feedback. We appreciate your efforts in reviewing our manuscript and are glad to hear that our responses addressed your concerns.

---

### Official Review · Reviewer_GLFg · 2025-07-01

**Clarity:** 3
**Significance:** 2
**Originality:** 2
**Rating:** 4
**Confidence:** 4

**Summary:**

FlexWorld is a framework for generating flexible-view 3D scene from a single input image. It maintains a persistent 3DGS representation that is progressively expanded: a video-to-video (V2V) diffusion model synthesizes novel view videos from incomplete renders, new 3D content is extracted via dense stereo depth estimation and integrated into the scene, and the combined representation is jointly optimized for visual and geometric fidelity.
By fine-tuning a video model on high-quality depth-aligned training pairs, FlexWorld achieves consistent novel view synthesis under large camera variations. Extensive experiments on RealEstate10K, Tanks-and-Temples, and DL3DV-10K demonstrate that FlexWorld outperforms prior video-based and image-based novel view synthesis methods in perceptual metrics camera accuracy, and single-image-to-3D rendering quality. FlexWorld also uniquely supports extrapolating existing 3D scenes into broader flexible-view environments.

**Questions:**

Please refer to Weaknesses.

**Ethical Concerns:**

["NO or VERY MINOR ethics concerns only"]

**Final Justification:**

After the rebuttal, most of my concerns are solved, so I raise my final rating.

**Limitations:**

yes

**Quality:**

2

**Strengths And Weaknesses:**

Strengths:
1. Maintains and refines a single 3DGS scene rather than rebuilding per iteration, enabling seamless expansion and scene extrapolation.

2. Combines geometry-aware scene integration with 3DGS optimization losses and optional image-level refinement, resulting in both visual polish and geometric accuracy.

3. Demonstrates superiority across multiple benchmarks and tasks—novel view synthesis, single-image 3D scene generation, and scene extrapolation—using a wide array of perceptual and geometric metrics.

Weaknesses:
1. Employing a video diffusion model to “paint” novel views based on point‐cloud renders is a well‐established approach in camera‐controlled video and 3D scene generation, so the core pipeline raises questions about its overall novelty.

2. The manuscript does not convincingly demonstrate why a 3DGS representation outperforms traditional point‐cloud outputs which can be directly obtained from methods like DUSt3R or VGGT.

3. Evaluation is centered on 2D metrics; adopting a 3D‐centric metric suite such as WorldScore would provide a more comprehensive assessment of camera‐controlled video generation and underlying scene quality.

4. I also want to know the accumulated error during progressively expanding the scene.

---

> ### Author Rebuttal · Authors · 2025-07-31
>
> We sincerely thank you for your time and careful review of our manuscript. Your insightful feedback has been invaluable in enhancing our work. We will carefully incorporate all of your suggestions in the revised manuscript. Our point-by-point responses to your comments follow below.
>
> #### **Weakness 1: Novelty of the Pipeline**
>
> We thank you for your feedback. We would like to respectfully clarify how our framework's architecture represents a significant and novel departure from established approaches.
>
> The core distinction lies in our use of a **persistent 3D scene**, as detailed in Sec 3.2. This allows us to leverage rich geometric and color information from the existing scene to guide subsequent expansions into a final, flexible-view 3D scene. This stateful, 3D-aware approach is a fundamental departure from frameworks like ViewCrafter, which **discard intermediate geometry after each step** and instead construct temporary 3D structures using only historical RGB information. This framework modification leads to superior final results, as demonstrated in Fig. 8 by the comparison between setups \(c\) and (d).
>
> We have conducted a new quantitative experiment inspired by your and reviewer v55n's suggestions to further validate the superiority of our framework design. We **carefully isolated the framework's contribution** by comparing two setups using the same V2V model and no image refinement: (1) a ViewCrafter-style framework and (2) our proposed framework. We generated 360° 3D scenes for 155 test cases (due to time constraints, we selected the first 5 out of 20 total categories and the first 31 out of 100 images from each, totaling 155 scenes for evaluation) under five key configurations, rendered 360° videos from these scenes, and then evaluated the videos using the official WorldScore protocol. The results below demonstrate that our framework architecture consistently outperforms the alternative across all relevant metrics.
>
> | Configuration                                            | WorldScore (Overall) | Camera Ctrl | Object Ctrl | Content Align | 3D Consist | Photo Consist | Style Consist | Subjective Qual |
> | -------------------------------------------------------- | -------------------- | ----------- | ----------- | ------------- | ---------- | ------------- | ------------- | ------------------ |
> | Our V2V+ ViewCrafter style's  framework (w/o refinement) | 49.97                | N/A         | 70.16       | 55.66         | 62.66      | 37.77         | 86.6          | 36.95              |
> | Our framework (w/o refinement)                           | 56.34                | N/A         | 73.06       | 58.3          | 71.07      | 46.62         | 96.48         | 48.84              |
>
> (Note: Higher is better for all metrics. The Camera Control metric is not applicable (N/A) as our final scene camera movements are more complex than the WorldScore defaults.)
>
> Furthermore, this persistent 3D structure uniquely endows our framework with the ability to perform **scene extrapolation**, as detailed in Sec. 4.4, allowing our model to seamlessly extend an existing 3D scene, **a capability not present in prior works like ViewCrafter or See3D**. We believe these supplementary explanations demonstrate that our framework design is, in itself, a novel and impactful contribution. We will add this quantitative study to the ablation section in our revised manuscript.
>
> #### **Weakness 2: Justification for Using 3DGS**
> Thank you for this important question. Our choice of 3D Gaussian Splatting (3DGS) over simpler point-cloud representations from methods like DUSt3R is motivated by two fundamental advantages essential to our framework: rendering quality and functional capability.
>
> First, regarding quality, a raw point cloud is merely a geometric scaffold. Rendering it directly often produces sparse results with holes and lacks photorealism. In contrast, 3DGS is a complete scene representation where each Gaussian splat models not just geometry but also local appearance, enabling the synthesis of high-fidelity, anti-aliased novel views that capture complex view-dependent effects. Furthermore, our method operates with known camera trajectories, making direct reconstruction into a high-fidelity 3DGS representation a more natural and effective choice compared to methods using unknown camera trajectories like VGGT and DUSt3R.
>
> Most critically, however, 3DGS is inseparable from our **progressive scene expansion** process. A 3DGS scene is composed of explicit, editable primitives. This property is foundational to our method, as it allows our framework to iteratively fuse new information and seamlessly expand the 3D scene. This dynamic, progressive construction is a core novelty of our work and is fundamentally different from the static, holistic point-cloud outputs generated by methods like DUSt3R.
>
> In summary, 3DGS is not merely a post-processing choice for better visuals; it is a foundational component that enables the core functionality of our method. We are very grateful for your suggestion which prompted this discussion, and we will add this explanation to the revised manuscript.
>
>
> #### **Weakness 3: Evaluation Metrics**
>
> We sincerely thank you for this excellent and highly constructive suggestion. Adopting a 3D-centric metric is indeed crucial for a comprehensive assessment. Following your advice, we have conducted a new evaluation using the official **WorldScore benchmark**.
>
> Due to the time constraint, we performed a head-to-head comparison between our method and the baseline, ViewCrafter, and evaluated a substantial subset of 300 scenes from the WorldScore static test dataset (2000 total). The subset was created by systematically selecting the first 15 scenes from each of the 20 categories to ensure diversity (including indoor/outdoor, stylized/photorealistic scenes). The evaluation followed the official WorldScore protocol, with the generation methods for both the test videos and 3D scenes remaining consistent with Sections 4.2 and 4.3 in the original paper, respectively. Due to the time limitation, we chose the first 100 scenes to do a 3D scene generation evaluation. The detailed results are presented in the table below (note: higher is better for all metrics):
>
> Quantitative comparison of **camera‐controlled video generation**:
> | Methods     | WorldScore (Overall) | Camera Ctrl | Object Ctrl | Content Align | 3D Consist | Photo Consist | Style Consist | Subjective Qual |
> | ----------- | ----------------- | ----------- | ----------- | ------------- | ---------- | ------------- | ------------- | ------------------ |
> | ViewCrafter | 66.81             | 78.25       | 48.92       | 50.63         | 81.88      | 74.9          | 68.05         | 65.01              |
> | Ours        | **68.37**             | 82.79       | 44.92       | 49.35         | 84.7       | 90.53         | 64.36         | 61.97              |
>
> Quantitative comparison of **underlying scene quality**:
> | Methods      | WorldScore (Overall) | Camera Ctrl | Object Ctrl | Content Align | 3D Consist | Photo Consist | Style Consist | Subjective Qual |
> | :---------- | :------------------: | :---------: | :---------: | :-----------: | :--------: | :-----------: | :-----------: | :--------------: |
> | ViewCrafter |         66.2         |    81.02    |    69.0     |     47.77     |   83.46    |     81.30     |     64.88     |      35.94       |
> | Ours        |      **67.65**       |    80.35    |    69.5     |     46.27     |   87.47    |     88.42     |     61.72     |      39.84       |
>
>
> The results from the WorldScore evaluation show that our model achieves better overall scores, which confirms its strong performance and aligns with the findings in Tables 1 and 2 of our original manuscript. Notably, our method still shows improvement even though the camera movement in the WorldScore benchmark is relatively simple. Moreover, our approach demonstrates a more significant advantage in scenarios with large camera variations, as illustrated in Figure 3 and 9 of the manuscript.
>
> In conclusion, this new, more comprehensive 3D-centric evaluation confirms the superiority of our method over the baseline. We will conduct these new experiments on the complete benchmark and add the results to the experimental section of our revised manuscript.
>
> #### **Weakness 4: Accumulated Error**
>
> Thank you for this critical question regarding error accumulation during progressive scene expansion. While it is common for accumulated errors to arise in multi-round iterative generation frameworks, our framework is specifically designed to mitigate this issue.
>
> As mentioned in our response to Weakness 1, the core of our approach is the maintenance of a **persistent 3D structure**. At each expansion step, the existing 3D scene (including its geometry and color) serves as a strong, coherent anchor for the generation of the next video segment. This ensures that newly generated content is well-aligned with the established scene, fundamentally reducing the kind of accumulated error that can occur in frameworks that lack this persistent geometric memory.
>
> By the way, we recognize that depth estimation inaccuracies may introduce accumulated errors, though improved estimators (e.g., VGGT) could reduce them. This limitation will be further discussed in the final manuscript.
>
>
> Thank you again for your constructive comments. We believe the revisions and additional experiments, undertaken in response to your feedback, have thoroughly addressed your concerns and will strengthen the overall quality of our manuscript.

---

> ### Comment · Reviewer_GLFg · 2025-08-01
>
> Thanks for your reply. I got some other questions.
> 1. Upon reviewing Table 1, I noticed that the results for PSNR, SSIM, and LPIPS are significantly inferior compared to those of Wonderland, which also utilized Cogvideo as their base model. What could be the reasons for this inconsistency?
>
> 2. How did you ensure scale consistency between the original pose in Re10K and the estimated point clouds? During inference, when given a single reference image, you should first build the initial point cloud and then render the video as conditioning. However, it seems that the scale of the estimated point cloud is considerably smaller than that of the ground truth pose.
>
> 3. Since worldScore do not provide ground-truth camera pose, how the camera control metric is computed?

---

> > ### Author Response · Authors · 2025-08-01
> >
> > We sincerely thank you for your continued engagement and for these insightful follow-up questions. We appreciate the opportunity to further clarify these important details.
> >
> > **Q1. Metric Discrepancy with Wonderland**
> >
> > Thank you for this sharp observation. We clarify that the **primary reason** for this discrepancy in PSNR, SSIM, and LPIPS scores is **the difference in the number of frames used for assessment**.
> >
> > As we state in our manuscript (Lines 208-209), our evaluation is performed on all generated frames of the video sequence. In contrast, the Wonderland paper explicitly mentions in their Section 4.1 (under "Evaluation Metrics - Visual Similarity") that they only use the first 14 generated frames for comparison against the ground truth.
> >
> > It is an inherent characteristic of generative models that diversity and potential deviations from the ground truth increase over longer generation. Consequently, evaluating a full sequence naturally results in lower average scores on reference-based metrics compared to evaluating only a short, initial segment.
> >
> > We chose to evaluate on all frames because, in a setting where different models may produce sequences of varying lengths, we believe this provides a fairer and more comprehensive comparison. Arbitrarily truncating all outputs to a fixed short length like 14 frames might not accurately reflect the overall quality of each method. **All comparisons within our paper were conducted under this unified and consistent protocol, ensuring their fairness.**
> >
> > Furthermore, a direct, fully controlled comparison with Wonderland is challenging as the **its model and evaluation code are not publicly available** at this time. We would be very willing to conduct such a comparison should their assets be released.
> >
> > **Q2. Scale Consistency**
> >
> > We sincerely apologize for the lack of clarity in our manuscript that led to this confusion. To clarify, on the RE10K dataset, we also utilized the MASt3R model to **re-annotate** the camera poses and did not use the poses from the original dataset. This process is consistent with the methodology described in Lines 202-203 of our paper and was done to ensure scale consistency.
> >
> > This choice was made to guarantee that our model and all baseline models were evaluated under identical and fair conditions, using a single, consistent camera data format. This approach prevents any potential issues arising from scale or coordinate system inconsistencies between different data sources.
> >
> > We acknowledge this was an omission in our original manuscript. We will revise the text to explicitly detail this processing step, thereby ensuring full transparency regarding our evaluation protocol.
> >
> > **Q3. Camera Control Metric in WorldScore**
> >
> > Thank you for your quetion. First, we would like to reaffirm that we adopted the WorldScore benchmark based on your valuable suggestion, and our evaluation is in strict adherence with the official protocol detailed in the WorldScore paper to ensure the validity of our results.
> >
> > In our supplementary quantitative comparison of camera-controlled video generation and underlying scene quality, the camera control metric is indeed computable. The calculation process is as follows:
> >
> > The WorldScore benchmark provides an image and a camera trajectory, which is sourced from its camera templates. It then tasks the generation model with creating a corresponding video or scene. To evaluate this, the trajectory of the output video is estimated using DROID-SLAM. After computationally aligning this estimated trajectory with the ground-truth (GT) trajectory, the resulting alignment error is used as the score. A lower error indicates more precise control over the camera movement.
> >
> > For a more detailed breakdown of the procedure and the specific computation formulas used, we recommend consulting the original WorldScore paper [1] to obtain the complete methodology.
> >
> > [1] Duan H, Yu H X, Chen S, et al. Worldscore: A unified evaluation benchmark for world generation[J]. arXiv preprint arXiv:2504.00983, 2025.
> >
> > Thank you again for your thoughtful questions and valuable feedback. Our responses have carefully addressed each of your concerns, and we hope these clarifications meet your expectations.

---

> > > ### Comment · Reviewer_GLFg · 2025-08-01
> > >
> > > As you mentioned, it is an inherent characteristic of generative models that diversity and potential deviations from the ground truth increase over longer generations. Does this mean that computing metrics on these pixels is meaningless? You also mentioned that different models may produce sequences of varying lengths, and you handle this by uniformly excluding cameras from the original trajectory to match the required length. Can this strategy ensure a fair comparison and how to understand this? For example, if ViewCrafter generates 25 frames while your method generates 49 frames, can you simply double the sampling stride to ensure the same movement range? In this case, computing metrics on the first 14 frames would be fair. However, I believe the best way to evaluate the metric is to compute it only on the visible pixels in the first frame, which is deterministic. Is this possible?
> > >
> > > Regarding the second question, I am wondering whether it is fair to recompute the ground-truth camera pose of the training data to match the scale of the estimated point cloud. It seems that the scale of the recomputed camera pose is much smaller than the original pose. Is it a fair comparison with previous Plücker-based video generation methods like CamCtrl that use the original pose?
> > >
> > > For the third question, since I am not familiar with the details of Worldscore, I noticed that for each image, they do not provide the camera trajectory (i.e., per-frame extrinsic and intrinsic parameters). I am curious about how to compute the metric without these, as you need these variables for rendering 3D-GS as conditioning.

---

> > > > ### Author Response · Authors · 2025-08-03
> > > >
> > > > Thank you for your continued discussion. We provide detailed responses to each of your points below.
> > > >
> > > > **Q1. On Evaluation Metrics**
> > > >
> > > > First and foremost, we wish to clarify that our work primarily focuses on generating long novel-view videos under **specific camera trajectories**, especially those with **large variations**, which is a significant aspect of 3D scene generation. Our evaluation protocol is designed to serve this goal; specifically, this strategy ensures that **all methods are evaluated over the exact same camera trajectory range**.
> > > >
> > > > We acknowledge the inherent limitations of metrics like PSNR, SSIM, and LPIPS for generative tasks, where divergence from the ground truth is expected. However, these remain **standard, established metrics** in the field. While their absolute values may be lower on challenging long-generation tasks, they still serve as **valuable indicators of perceptual quality and consistency**. Therefore, **we believe our original evaluation, which employed a comprehensive set of seven distinct metrics (including all those used by ViewCrafter and Wonderland)** and incorporated your previous suggestion of WorldScore, was already thorough in demonstrating our method's effectiveness.
> > > >
> > > > Nevertheless, to directly address your concern and further validate our claims, we have conducted an additional experiment comparing performance on a shorter sequence of 14 frames, as you suggested. The results, shown below, confirm that our method still outperforms ViewCrafter and achieves performance comparable to that reported in the original Wonderland paper (noting a direct numerical comparison is not possible as their code and model is not public, preventing evaluation on identical data).
> > > >
> > > > |Models| PSNR ↑| SSIM ↑|LPIPS ↓|
> > > > |-|-|-|-|
> > > > |ViewCrafter (14 frames)|17.82|0.659|0.283|
> > > > |**Ours (14 frames)**|**19.38**|**0.681**|**0.236**|
> > > >
> > > > Regarding your proposal to evaluate only on the visible pixels from the first frame, we have two primary concerns. To our knowledge, no standard benchmarks provide the direct necessary per-pixel visibility ground truth for subsequent frames, making this metric difficult to implement and evaluate. More importantly, such a metric would be insufficient for evaluating the task of novel view synthesis. It would fail to assess the geometric consistency or realism of the generated views. For instance, a model could "cheat" by perfectly reprojecting visible pixels from the first frame and rendering blackness for all other areas.
> > > >
> > > > In summary, Our experimental setup was designed to address the problem of novel-view video generation under large camera variations, and our comprehensive metrics have validated our claims. We are always open to adopting new, well-motivated evaluation protocols as they become established in the community.
> > > >
> > > > **Q2. On Scale Consistency and Fairness**
> > > >
> > > > To further clarify our methodology, for plücker-based methods (i.e., MotionCtrl, CameraCtrl), we used the original RE10K camera poses as input. For See3D, ViewCrafter, and Ours, we used MASt3R to estimate the initial point cloud and use its estimated camera for rendering incomplete input video, creating an appropriate input for them. We also wish to clarify that **our primary baselines are ViewCrafter and See3D, and all comparisons were conducted under identical conditions**. We included CameraCtrl due to its relevance and made every effort to align our experimental evaluations. To clarify, our previous statement about 'utilizing the MASt3R model to re-annotate the camera poses' refers specifically to the computation of the final camera error metrics (R_err, T_err) that estimate cameras from the GT videos and the generated videos. We will revise the tables in our manuscript to explicitly label which camera poses were used for input vs. evaluation (e.g., "Original" / "Re-estimated") and modified relative expression in the text.
> > > >
> > > > Furthermore, we clarify that **we have never claimed that "the scale of the recomputed camera pose is much smaller than the original pose,"** nor have our experiments observed this phenomenon in our setting. The experimental evidence is as follows (note: the scale is estimated by averaging the distance between the two farthest camera centers for each scene in our evaluation dataset.):
> > > >
> > > > |Source|Scale|
> > > > |-|-|
> > > > |MASt3R| 1.56 |
> > > > |Original| 1.66 |
> > > >
> > > > In summary, we have made our best effort to conduct a fair and thorough evaluation. If you have any further suggestions for improving our protocol, we are very willing to implement them.
> > > >
> > > > **Q3. On Obtaining Camera Trajectories from WorldScore**
> > > >
> > > > WorldScore **do provide the camera trajectory** for each image. WorldScore defines a set of camera motion templates, such as `move_left`, `move_right`, etc. The official benchmark code then uses the `CameraGen` class (from `worldscore/benchmark/helpers/camera_generator.py`) to convert these semantic templates into **camera trajectories with per-frame extrinsic and intrinsic parameters** for our evaluation.

---

> > > > > ### Comment · Reviewer_GLFg · 2025-08-05
> > > > >
> > > > > I am not questioning the experimental setup, and I mean no harm. I am simply considering ways to make the evaluation more reasonable to benefit the community.
> > > > >
> > > > > For the first question, I acknowledge that PSNR, SSIM, and LPIPS are standard metrics in the field and they can evaluate both camera controllability and novel view synthesis by comparing generated video frames with ground-truth ones. However, in my opinion, novel view synthesis evaluation requires aligned generated frames and ground-truth ones. Due to the randomness of generation, it is impossible to generate the same content as ground-truth videos except for zoom-in camera trajectory. Hence, I think these metrics are more suited to evaluating camera controllability.
> > > > >
> > > > > In such a case, the most reasonable approach would be to compute PSNR on seen pixels in the first frame. I understand that there currently does not exist such a benchmark, but videos with camera pose inherently have a 3D structure. If the depth is estimated, we can obtain the visible pixels by projection. I do not require the authors to perform such experiments (current experiment results are enough for me, I acknowledge the efforts); I just want to discuss whether such a metric would be more reasonable.
> > > > >
> > > > > For the third question, I am curious about the scale of the camera pose generated by WorldScore and whether it might be misaligned with that of MASt3R. For example, if MASt3R estimates the point clouds at a small scale while WorldScore generates extrinsics with large camera movements, how would this affect the final score?

---

> > > > > > ### Author Response · Authors · 2025-08-05
> > > > > >
> > > > > > Thank you for clarifying and for the spirit of this discussion. We genuinely appreciate this exchange of ideas and are happy to delve deeper into these important topics with you.
> > > > > >
> > > > > > **On Metrics for Novel View Synthesis**
> > > > > >
> > > > > > Regarding your proposal to evaluate only on the visible pixels from the first frame, we agree that this is an insightful suggestion for a supplementary analysis of reconstruction fidelity and consistency. As we continue this discussion, we would like to build on our previous points and explore further for such a metric.
> > > > > >
> > > > > > As we mentioned, a primary concern is that the metric only reflects how well the model **preserves** the old scene but not how well it **generates** consistent new scenes. This leads to a potential "cheating" failure mode: any image inpainting model that simply copies the known region will hit the highest score, but the generated inpainted region may be inconsistent and fail the 3D reconstruction task. Furthermore, this metric would face challenges in correctly evaluating **occlusion**. For example, consider a camera pulling back from a room. A good generative model might synthesize a previously occluded chair. Under the proposed metric, the pixels corresponding to the new chair would be penalized because they do not match the background that was visible in the initial frame's projection, even though generating the chair is a correct and desirable outcome.
> > > > > >
> > > > > > Furthermore, as you rightly point out, while one could potentially obtain the visible pixels via projection using a depth estimation method, that estimation method would need to be exceptionally robust; otherwise, it would introduce additional sources of error into the evaluation.
> > > > > >
> > > > > > In summary, we agree that this is a thought-provoking direction for future metric development. However, we believe it requires further research to address these challenges before it could be widely adopted as a robust standard. We are always open to adopting new, well-motivated evaluation protocols as they become established in the community.
> > > > > >
> > > > > >
> > > > > > **On the Camera Pose of WorldScore**
> > > > > >
> > > > > > You are right to consider the potential scale mismatch. The WorldScore benchmark framework accounts for this directly with a controllable parameter named `camera_speed`, which adjusts the magnitude of the camera's scale. This value varies between models.
> > > > > >
> > > > > > We calibrated this parameter to 0.1 for our experiments. This value ensures that the camera movement is appropriately scaled relative to the scene's content, creating a motion effect consistent with the official WorldScore examples and ensuring a meaningful evaluation.

---

> > > > > > > ### Comment · Reviewer_GLFg · 2025-08-06
> > > > > > >
> > > > > > > Thanks for your response. Most of my concerns have been resolved.
> > > > > > >
> > > > > > > I have one last question: Is the camera speed set to 0.1 the default setting in WorldScore, and do all compared methods share the same setting?

---

> > > > > > > > ### Author Response · Authors · 2025-08-06
> > > > > > > >
> > > > > > > > We are pleased we could address most of your concerns. In response to your final question, WorldScore does not have a universal default setting; the `camera_speed` parameter varies across models (e.g., 0.0005 in WonderJourney, 0.002 in WonderWorld, and 2 in LucidDreamer, with more examples in the official code `config/model_configs`). For the new WorldScore comparison we provided during this rebuttal, both our model and ViewCrafter were benchmarked under the same setting. Thank you again for your feedback.

---

### Official Review · Reviewer_v55n · 2025-07-02

**Clarity:** 4
**Significance:** 3
**Originality:** 2
**Rating:** 5
**Confidence:** 3

**Summary:**

FlexWorld presents a comprehensive framework for 3D scene synthesis under large view-point changes in 360 degrees. Powering FlexWorld is a fine-tuned V2V model and a progressive scene expansion and integration framework.

**Questions:**

- What are the quantitative results for ablations in Figure 8? It would be good to see the quantitative results w/o FLUX refinement and after FLUX refinement. This would be especially helpful if metrics for strong baseline ViewCrafter is included.
- How are unobserved regions (black regions in figure 6) from training dataset creation handled during V2V training?
- Are there more details about any filtering done on DL3DV-10K dataset and the camera trajectories used to rendering video sequences?

**Ethical Concerns:**

["NO or VERY MINOR ethics concerns only"]

**Final Justification:**

Authors addressed my concerns regarding using complicated super resolution and refinement (flux) in their pipeline. The ablation shows that the core of the method, V2V on new dataset, yields the most significant contribution. I agree with reviewers that the V2V is not novel and dataset pipeline is the main novelty. The evaluations are thorough and the dataset pipeline is also well documented.

**Limitations:**

Yes

**Quality:**

4

**Strengths And Weaknesses:**

Strengths

- The methods are described clearly and experiments show high quality of work.
- Strong quantitative results on both novel view synthesis and 3D scene generation on unseen test sets against baselines.
- Strong qualitative results on novel view synthesis.
- Proposed methods, including training data construction and generation framework, are principled and well motivated, with qualitative ablation on inclusion of video diffusion, zooming on camera trajectory, and refinement.

Weaknesses

- The paper mentions the stronger V2V model is mainly driven by better construction of training data (see Figure 6) and a vastly stronger base V2V model, CogVideoX-5B-I2V. However, the technical novelty behind this remain rather limited.
- Full pipeline is rather complex, requiring v2V, Real-ESRGAN for super resolution, and FLUX for image refinement. Contributions of individual components are unclear.
- Lack of quantitative ablation, specifically on refinement using a powerful image model, FLUX.1-dev, which may impact the quantitative comparisons.

---

> ### Author Rebuttal · Authors · 2025-07-31
>
> We sincerely appreciate the time and effort you have devoted to reviewing our manuscript and providing insightful feedback. Your thoughtful comments have significantly contributed to improving the quality of our work. We have thoroughly addressed each of your suggestions and will implement the necessary revisions in the final version of the manuscript. Below, we provide detailed responses to each of your points.
>
> #### **Weakness 1: Limited Technical Novelty**
>
> We thank you for your feedback. While we agree that our method leverages a strong base model and an effective data construction strategy, we would like to respectfully clarify that our core technical novelty lies in the architectural design of our generation framework itself.
>
> Our primary contribution is a novel framework that maintains and iteratively refines a **persistent 3D structure**. (as detailed in Lines 146-148). This allows us to leverage rich geometric and color information from the existing scene to guide subsequent expansions into a final, flexible-view 3D scene. This stateful, 3D-aware approach is a fundamental departure from frameworks like ViewCrafter, which discard intermediate geometry after each step and instead construct temporary 3D structures using only historical RGB information. This framework modification leads to superior final results, as demonstrated in Fig. 8 by the comparison between setups \(c\) and (d), and is also supported by the table in the ablation study conducted at your suggestion (presented in the response to Weaknesses 2 & 3 & Question 1). Furthermore, this persistent structure enables **a unique capability not present in previous methods: 3D scene extrapolation, allowing our model to seamlessly extend a scene beyond its initial boundaries** (Sec. 4.4).
>
> We believe these supplementary explanations demonstrate that our framework design is, in itself, a novel and impactful contribution. We will add this new quantitative study and its corresponding analysis to our revised manuscript.
>
>
> #### **Weaknesses 2 & 3 & Question 1: Pipeline Complexity and Lack of Quantitative Ablation**
>
> We sincerely thank the reviewer for this insightful and constructive feedback. We agree that a quantitative ablation study is crucial for clarifying the contributions of each component in our framework and for providing a clearer comparison with strong baselines (i.e., ViewCrafter).
>
> First, for the sake of clarity, we wish to address the concern about the refinement module's impact on our main quantitative comparisons. As stated in the original manuscript (Lines 229-231), the results in Table 2 follow the experimental protocol of ViewCrafter, where metrics are calculated from 3D scenes reconstructed from videos generated *without* FLUX refinement. Therefore, the use of FLUX for refining the 360° flexible-view 3D scene shown in Fig. 1b does not influence those specific results.
>
> Nevertheless, we fully agree that a quantitative ablation study is crucial for delineating the contribution of each component. Following your suggestion, we have conducted a new quantitative ablation study. Measuring the quality of 360° scenes is challenging; fortunately, a recently released **WorldScore** benchmark [1] (April 2025, also suggested by Reviewer GLFg) provides a way for this. For this study, we generated 360° 3D scenes for 155 test cases (due to time constraints, we selected the first 5 out of 20 total categories and the first 31 out of 100 images from each, totaling 155 scenes for evaluation) under five key configurations, rendered 360° videos from these scenes, and then evaluated the videos using the official WorldScore protocol. The five configurations are:
> 1.  ViewCrafter's V2V Model + Our Framework (with super-resolution, without refinement)
> 2.  Our V2V Model + ViewCrafter's Framework (with super-resolution, without refinement)
> 3.  Our Framework (without super-resolution or refinement)
> 4.  Our Framework (with super-resolution, but without refinement)
> 5.  Our Full Framework (with super-resolution and refinement)
>
> The results are presented below. (Note: Higher is better for all metrics. The Camera Control metric is not applicable (N/A) as our scenes involve complex camera paths.)
>
> | Configuration                                                | WorldScore (Overall) | Camera Ctrl | Object Ctrl | Content Align | 3D Consist | Photo Consist | Style Consist | Subjective Qual |
> | ------------------------------------------------------------ | -------------------- | ----------- | ----------- | ------------- | ---------- | ------------- | ------------- | ------------------ |
> | (1) Viewcrafter's V2V+ our framework (w/o refinement)        | 44.12                | N/A         | 65.81       | 39.93         | 61.4       | 0.0           | 89.4          | 52.33              |
> | (2) Our V2V+ ViewCrafter style's  framework (w/o refinement) | 49.97                | N/A         | 70.16       | 55.66         | 62.66      | 37.77         | 86.6          | 36.95              |
> | (3) Our framework (w/o super-resolution, w/o refinement)     | 56.55                | N/A         | 72.74       | 57.25         | 69.85      | 51.72         | 96.3          | 47.96              |
> | (4) Our framework (w/o refinement)                           | 56.34                | N/A         | 73.06       | 58.3          | 71.07      | 46.62         | 96.48         | 48.84              |
> | (5) Our full framework (w/ refinement)                       | 55.6                 | N/A         | 71.29       | 57.09         | 70.81      | 45.69         | 96.66         | 47.67              |
>
>
> This comprehensive ablation provides several key insights that quantitatively support some of the qualitative results in Figure 8:
>
> * **Impact of our V2V Model**: Comparing **(1)** and **(4)** demonstrates the superiority of our trained V2V model. It significantly boosts performance across nearly all metrics, which aligns with the qualitative improvements shown between Fig. 8a and Fig. 8d.
> * **Impact of our Framework**: Comparing **(2)** and **(4)** highlights the effectiveness of our core framework design. When both use our V2V model, our framework substantially outperforms the ViewCrafter-style framework, confirming the observations from Fig. 8c and Fig. 8d and supporting the claim in the responses to Weakness 1.
> * **Impact of Super-Resolution**: The difference between **(3)** and **(4)** is minimal, indicating that the Real-ESRGAN super-resolution component is an optional step for potentially enhancing visual detail and not a primary contributor to our framework.
> * **Impact of FLUX Refinement**: Finally, comparing **(4)** and **(5)** directly addresses your question about the refinement step. As we elaborate in the Appendix (Line 492), the refinement step is used solely to enhance the visual quality of the generated scenes (which is consistent with the qualitative results in Fig. 8d and 8e), and it even leads to a slight decrease in comprehensive 3D metrics like WorldScore.
>
> We believe this new study thoroughly clarifies the contribution of each component and quantitatively validates the ablations presented in Figure 8. We will incorporate a full quantitative ablation study and our analysis into the "Ablation study" subsection in the revised manuscript.
>
> [1] Duan H, Yu H X, Chen S, et al. Worldscore: A unified evaluation benchmark for world generation[J]. arXiv preprint arXiv:2504.00983, 2025.
>
>
> #### **Question 2: Handling of Unobserved Regions**
>
> Thank you for this insightful question regarding our training data. In our current implementation, we do not apply any special handling (e.g., masking or inpainting) for the black, unobserved regions in the rendered training videos. This design choice was made to simplify the data framework and use a standard I2V model for our V2V task with minimal architectural changes. The model learns to interpret these unobserved areas as part of the input condition alongside the visible scene content. We will add sentences to the implementation details section of our manuscript to clarify this point.
>
> #### **Question 3: Dataset Filtering and Camera Trajectories**
>
> Thank you for asking for these important details.
>
> **Dataset Filtering:** We performed a rigorous, multi-step filtering process on the DL3DV-10K dataset. First, we discarded scenes with low image resolutions (below 540x960) or incomplete camera annotations. Next, as a critical quality control step, we verified consistency between the provided DL3DV camera parameters and those from a COLMAP reconstruction, removing any samples with significant discrepancies. This yielded our final training set of 10,253 high-quality scenes.
>
> **Camera Trajectories:** The trajectory choice was straightforward. As described in the manuscript, for each scene, we simply sample one continuous camera trajectory of a fixed length (49 frames) at random from the scene's original sequence.
>
> We will add these explicit details to the revised manuscript. To fully support reproducibility, we also affirm our commitment to releasing the data construction code upon the paper's publication.
>
> We are truly grateful for your time and expertise in reviewing our work. Your comments were invaluable in guiding our revisions, and we hope that the additional clarifications and experiments now fully address your concerns while improving the overall quality of the paper.

---

> > ### Comment · Reviewer_v55n · 2025-08-04
> > **Re: Rebuttal by Authors**
> >
> > Thanks for the detailed response by authors.
> >
> > The ablation on configurations are helpful in showing the impacts of various parts of the pipeline. The method appears solid without super resolution or flux refinement.

---

> > > ### Author Response · Authors · 2025-08-04
> > >
> > > Thank you for your positive feedback. We are glad that the ablation studies were helpful and truly appreciate your recognition of our method's solidness. Your valuable comments have significantly helped us improve our paper. Thanks again!

---

### Official Review · Reviewer_wGjX · 2025-07-03

**Clarity:** 4
**Significance:** 2
**Originality:** 2
**Rating:** 4
**Confidence:** 4

**Summary:**

The paper proposes FlexWorld, a progressive framework for generating 3D scenes that support flexible-view exploration from single images. FlexWorld uses an accumulated 3D Gaussian splatting representation progressively expanded by synthesizing new 3D content via a fine-tuned video-to-video (V2V) diffusion model. Extensive evaluations demonstrate improvements over existing methods in terms of visual quality, viewpoint flexibility, and consistency.

**Questions:**

1. Can the authors clearly explain the difference between the "3D Scene Generation" task and the "Novel View Synthesis" task in the context of this work? The current manuscript does not clearly differentiate these tasks. I don't find any difference.

2. Could the authors provide the exact number of samples included in the training sets? Specifically, how many 3DGS scenes from DL3DV were successfully reconstructed and utilized during training?


3. How much computation resource do you need to fit all Gaussians, apart from training the V2V model?

**Ethical Concerns:**

["NO or VERY MINOR ethics concerns only"]

**Final Justification:**

Most of my concerns have been addressed. I appreciate the efforts in preparing the dataset. It would be great if the authors have the plan to release them for public access.

I do not further raise my rating because of the novelty concern.

**Limitations:**

Yes

**Quality:**

3

**Strengths And Weaknesses:**

Strengths:
1. The motivation of the studied problem and the proposed method is clear.

2. The proposed method achieves better results compared to existing methods on Rel10k and Tanks-and-Temples datasets.

3. The proposed method is a system level integration of a custom V2V model, 3DGS buffer, and I2I refinement with  FLUX.1-dev.


Weakness:
1. Incremental novelty: The core technical novelty primarily relies on integrating existing concepts (3D Gaussian splatting and V2V diffusion). Although well-executed, individual components are incremental improvements over recent work like ViewCrafter or See3D. I appreciate the efforts in preparing incomplete-view datasets, i.e., training thousands of 3DGS on the DL3DV dataset to obtain paired training data, but I have to acknowledge the limited novelty of the overall framework.

2. The proposed method is very computationally demanding, requiring 30 minutes for generating a single scene. This limits its broader applicability.

---

> ### Author Rebuttal · Authors · 2025-07-31
>
> We would like to express our sincere gratitude for the valuable time you have dedicated to the peer review process and for your thorough examination of our work. Your constructive comments have been instrumental in enhancing the quality of our manuscript. We have carefully considered all suggestions and will incorporate the corresponding revisions in the final version of our manuscript. Below is our point-by-point response to your comments.
>
> #### **Weakness 1: Incremental Novelty**
>
> We sincerely thank you for your insightful feedback and for acknowledging the effort involved in our data preparation. We appreciate this opportunity to clarify our core novelty, which lies not only in combining existing components but more importantly, in the architectural design of our scene generation framework and the unique capabilities it enables.
>
> First, our primary contribution is a novel framework centered around a persistent 3D structure. This allows us to **leverage rich geometric and color information from the existing scene** to guide subsequent expansions into a final, flexible-view 3D scene. This stateful, 3D-aware approach is a fundamental departure from frameworks like ViewCrafter, which **discard intermediate geometry after each step** and instead construct temporary 3D structures using only historical RGB information. Our framework leads to superior final results, as demonstrated in Fig. 8 by the comparison between setups \(c\) and (d).
>
> We have conducted a new quantitative experiment to further validate the effectiveness of our main contribution, as suggested by reviewers v55n and GLFg. We **carefully isolated the framework's contribution** by comparing two setups using the same V2V model and no image refinement: (1) a ViewCrafter-style framework and (2) our proposed framework. We generated 360° 3D scenes for 155 test cases (due to time constraints, we selected the first 5 out of 20 total categories and the first 31 out of 100 images from each, totaling 155 scenes for evaluation) under five key configurations, rendered 360° videos from these scenes, and then evaluated the videos using the official WorldScore protocol. The results below demonstrate that our framework architecture consistently outperforms the alternative across all relevant metrics.
>
>
> | Configuration                                            | WorldScore (Overall) | Camera Ctrl | Object Ctrl | Content Align | 3D Consist | Photo Consist | Style Consist | Subjective Qual |
> | -------------------------------------------------------- | -------------------- | ----------- | ----------- | ------------- | ---------- | ------------- | ------------- | ------------------ |
> | Our V2V+ ViewCrafter style's  framework (w/o refinement) | 49.97                | N/A         | 70.16       | 55.66         | 62.66      | 37.77         | 86.6          | 36.95              |
> | Our framework (w/o refinement)                           | 56.34                | N/A         | 73.06       | 58.3          | 71.07      | 46.62         | 96.48         | 48.84              |
>
>
> (Note: Higher is better for all metrics. The Camera Control metric is not applicable (N/A) as our final scene camera movements are more complex than the WorldScore defaults.)
>
> Furthermore, this persistent 3D structure uniquely endows our framework with the ability to perform **scene extrapolation**, as detailed in Sec. 4.4, allowing our model to seamlessly extend an existing 3D scene, **a capability not present in prior works like ViewCrafter or See3D**. We believe these supplementary explanations demonstrate that our framework design is, in itself, a novel and impactful contribution. We will add this quantitative study to the ablation section in our revised manuscript.
>
> #### **Weakness 2: Computational Cost**
>
> We thank the reviewer for this important point. While the 30-minute computation time per scene is a current limitation, our method's runtime for generating a 3D scene is comparable to that of ViewCrafter under the same settings.
>
> Moreover, this issue will be addressed in future work without altering the core components of our approach. Our analysis indicates the primary bottlenecks are the video diffusion model's generation time and the iterative 3D Gaussian Splatting optimization. However, this can be substantially mitigated by leveraging rapid advancements in foundational models. For instance, recent progress in video model distillation [1] is dramatically reducing inference time, while emerging feed-forward 3DGS methods [2] can replace the costly optimization step. These technologies can be integrated into our framework in the future to make it significantly more efficient. To ensure full transparency, we will add a detailed discussion of this limitation and these potential future improvements to the "Limitations" section of our revised manuscript.
>
> [1] Yin T, Zhang Q, Zhang R, et al. From slow bidirectional to fast causal video generators[J]. arXiv e-prints, 2024: arXiv: 2412.07772.
>
> [2]Jiang L, Mao Y, Xu L, et al. AnySplat: Feed-forward 3D Gaussian Splatting from Unconstrained Views[J]. arXiv preprint arXiv:2505.23716, 2025.
>
> #### **Question 1: "3D Scene Generation" vs. "Novel View Synthesis"**
>
> We thank the reviewer for highlighting this lack of clarity and apologize for the confusion. We will explicitly define these terms in our final manuscript. In our work, we use them to distinguish two sequential stages:
>
> 1.  **Novel View Synthesis (NVS):** This refers specifically to the output of the video diffusion model, which generates a video clip simulating camera movement through a scene. It is fundamentally a 2D video generation task, creating a sequence of new views.
> 2.  **3D Scene Generation:** This refers to the subsequent process where we use the generated video to construct an explicit and renderable 3D scene representation (i.e., 3D Gaussian Splatting). This final 3D asset can be rendered from any arbitrary viewpoint.
>
> This distinction is adopted from the influential ViewCrafter paper [3], where our "Novel View Synthesis" is analogous to their video generation step (Fig. 3), and our "3D Scene Generation" corresponds to their "scene reconstruction" stage (Fig. 4). We will add these clarifications to the introduction of our revised manuscript.
>
> [3] Yu W, Xing J, Yuan L, et al. Viewcrafter: Taming video diffusion models for high-fidelity novel view synthesis[J]. arXiv preprint arXiv:2409.02048, 2024.
>
> #### **Question 2: Number of Training Samples**
>
> Thank you for this question. We trained our model on **10,253** successfully reconstructed 3D scenes from the DL3DV dataset, a widely-used collection of 3D data. This number was reached after a rigorous filtering process where we first excluded scenes with low image resolution (< 540x960) or missing camera parameters. Subsequently, we cross-validated the camera poses between the official DL3DV annotations and a COLMAP reconstruction cache, discarding any scenes with significant discrepancies to ensure high-quality data. We will update the manuscript to include these specific details about our dataset preparation. To further enhance reproducibility, we also commit to making our data processing scripts publicly available upon publication.
>
>
> #### **Question 3: Computational Cost for Fitting Gaussians**
>
> Thank you for your question regarding the reconstruction cost. For each of the 10,253 scenes in our training set, we fit the 3D Gaussians using the official implementation, running the optimization for 7,000 steps. This process takes approximately **5 minutes per scene** on a single NVIDIA A800 GPU. By parallelizing the workload on a server equipped with 8 NVIDIA A800 GPUs, we were able to complete the full dataset reconstruction in approximately 4-5 days. We will add these computational specifics to the implementation details section of our manuscript for transparency and reproducibility.
>
>
> Thank you again for your valuable comments and suggestions. We believe that the clarifications and supplementary experiments, made in response to your feedback, have appropriately addressed your concerns and further strengthened the persuasiveness of our paper.

---

> > ### Comment · Reviewer_wGjX · 2025-08-05
> >
> > I thank the authors for their detailed response. Most of my concerns have been addressed. I appreciate the efforts in preparing the dataset. It would be great if the authors have the plan to release them for public access.
> >
> > I currently keep my original rating. I will take the rebuttal into account as I finalize my review during the AC-reviewer rebuttal phase.

---

> > > ### Author Response · Authors · 2025-08-06
> > >
> > > Thank you for your feedback. We are glad to hear that most of your concerns have been addressed, and we will open-source the data processing pipeline upon publication. Please feel free to reach out if you have any further questions.

---

### Comment · Area_Chair_R23W · 2025-08-03
**Reminder: Discussion and Final Rating Update**

Dear Reviewers,

As we are now midway through the discussion phase, I would like to kindly remind you to review the authors' rebuttal and participate in the discussion. Please also update your review with a final rating accordingly.

Thank you very much for your time and valuable contributions to the review process.

Best regards,

Area Chair

---

### Note · Authors · 2025-08-12

We sincerely thank the Area Chair and all reviewers for the diligent, insightful, and highly constructive review process. The detailed rebuttal and discussions have been invaluable, and we are grateful for the opportunity to clarify our work. We are pleased that we were able to **resolve most of the reviewers' concerns** regarding novelty, evaluation, and specific implementation details.

To summarize our primary contribution: we propose a novel 3D generation framework centered around a **persistent 3D structure**, leveraging rich geometric and color information from the existing scene to guide subsequent expansions into a final, flexible-view 3D scene. To further substantiate our claims and directly address the feedback on evaluation, we undertook significant additional experiments during the rebuttal period. Most notably, we conducted **several new, targeted quantitative experiments**, including detailed ablation studies and a comparison on a 3D-centric metric, WorldScore. These new results provide strong evidence to clarify the contributions of each component and our method's superiority. **We commit that all clarifications made and necessary new experiments conducted during the rebuttal and discussion will be integrated into our final manuscript.**

We are confident that our detailed discussion and the corresponding revisions will significantly strengthen our paper. We respectfully hope that our clarifications and the compelling new evidence will be taken into positive consideration during the final decision-making process. Thank you again for your time and expertise.

---

### Decision · Program_Chairs · 2025-09-17

**Decision:**

Accept (poster)

**Comment:**

The paper presents a solid method with thorough experimental validation and well-prepared datasets. Reviewers acknowledge that most concerns have been addressed, and ablations demonstrate the impact of key pipeline components. Standard metrics are sufficient to evaluate camera controllability and novel view synthesis, and the method is robust without requiring additional components. Overall, the paper makes a meaningful contribution, and I therefore recommend acceptance.